# Finding any Waldo with zero-shot invariant and efficient visual search

Mengmi Zhang [1,2,3,4], Jiashi Feng[3], Keng Teck Ma[5], Joo Hwee Lim[4], Qi Zhao[6] & Gabriel Kreiman[1]

Searching for a target object in a cluttered scene constitutes a fundamental challenge in daily vision. Visual search must be selective enough to discriminate the target from distractors, invariant to changes in the appearance of the target, efficient to avoid exhaustive exploration of the image, and must generalize to locate novel target objects with zero-shot training. Previous work on visual search has focused on searching for perfect matches of a target after extensive category-specific training. Here, we show for the first time that humans can efficiently and invariantly search for natural objects in complex scenes. To gain insight into the mechanisms that guide visual search, we propose a biologically inspired computational model that can locate targets without exhaustive sampling and which can generalize to novel objects. The model provides an approximation to the mechanisms integrating bottom-up and top-down signals during search in natural scenes.

[1] Children's Hospital, Harvard Medical School, Boston, MA 02115, USA. [2] Graduate School for Integrative Sciences and Engineering, National University of Singapore, Singapore 138632, Singapore. [3] Department of Electrical and Computer Engineering, National University of Singapore, Singapore 138632, Singapore. [4] Visual Intelligence Unit, Image/Video Analytics Dept, A*STAR, Singapore 138632, Singapore. [5] Artificial Intelligence Program, Agency for Science, Technology and Research, Singapore 138632, Singapore. [6] Department of Computer Science and Engineering, University of Minnesota Twin Cities, Minneapolis, MN 55455, USA. Correspondence and requests for materials should be addressed to G.K. (email: gabriel.kreiman@tch.harvard.edu)

Visual search constitutes a ubiquitous challenge in natural vision, including daily tasks such as looking for the car keys at home. Localizing a target object in a complex scene is also important for many applications including navigation and clinical image analysis. Visual search must fulfill four key properties: (1) selectivity (to distinguish the target from distractors in a cluttered scene), (2) invariance (to localize the target despite changes in its appearance or even in cases when the target appearance is only partially defined), (3) efficiency (to localize the target as fast as possible, without exhaustive sampling), and (4) zero-shot training (to generalize to finding novel targets despite minimal or zero prior exposure to them).

Visual search is a computationally difficult task due to the myriad possible variations of the target and the complexity of the visual scene. Under most visual search conditions, the observer does not seek an identical match to the target object at the pixel level. The target object can vary in rotation, scale, color, illumination, occlusion, and other transformations. Additionally, the observer may be looking for any exemplar from a generic category (e.g., looking for any chair, rather than a specific one). Robustness to object transformations has been a fundamental challenge in the development of visual recognition models where it is necessary to identify objects in a way that is largely invariant to pixel-level changes (e.g., refs. [1–8], among many others). The critical constraint of invariance in recognition has led to hierarchical models that progressively build transformation-tolerant features that are useful for selective object identification.

In contrast with the development of such bottom-up recognition models, less attention has been devoted to the problem of invariance in visual search. A large body of behavioral[9,12] and neurophysiological[13–16] visual search experiments has focused on situations that involve identical target search. In those experiments, the exact appearance of the target object is perfectly well defined in each trial (e.g., searching for a tilted red bar, or searching for an identical match to a photograph of car keys). Some investigators have examined the ability to search for faces rotated with respect to a canonical viewpoint[17], but there was no ambiguity in the target appearance, therefore circumventing the critical challenge in invariant visual search. In hybrid search studies, the observer looks for two or more objects, but the appearance of those objects is fixed[18]. Several studies have evaluated reaction times during visual search for generic categories as a function of the number of distractors[19,20].

Template-matching algorithms perform poorly in invariant object recognition. In visual search, template-matching shows selectivity to distinguish a target from distractors, but fails to robustly find transformed versions of the target. Computer vision investigators have developed object detection and image retrieval approaches to robustly localize objects, at the expense of having to extensively train those models with the sought targets and exhaustively scan the image through sliding windows[21–25].

Most of these computer vision approaches bear no resemblance to the neurophysiological architecture of visual search mechanisms in cortex. In contrast with heuristic algorithms based on sequential scanning and class-specific supervised training, when presented with a visual search task, observers rapidly move their eyes in a task-dependent manner to search for the target, even when the exact appearance of the target is unknown and even after merely single-trial exposure to the target. When presented with an image, and before taking into account any task constraints, certain parts of the image automatically attract attention due to bottom-up saliency effects[26]. Task goals, such as the sought target in a visual search paradigm, influence attention allocation and eye movements at the behavioral[9,12,27,28] and neurophysiological levels[14,15,29,30]. Task-dependent modulation of neurophysiological responses is likely to originate in frontal cortical structures[15,31] projecting in a top-down fashion onto visual cortex structures[29,32]. Several computational models have been developed to describe visual search behavior or the modulation of responses in visual cortex during feature-based attention or visual search (e.g., refs. [10–12,27,33–39]).

Here, we quantitatively assess human visual search behavior by evaluating selectivity to targets versus distractors, tolerance to target shape changes, and efficiency. We conduct three increasingly more complex tasks where we measure eye movements while subjects search for target objects. To gain insight into the mechanisms that guide visual search behavior, we develop a biologically inspired computational model, and evaluate the model in terms of the four key properties of visual search. We show that humans can efficiently locate target objects despite large changes in their appearance and despite having had no prior experience with those objects. The computational model can efficiently localize target objects amidst distractors in complex scenes, can tolerate large changes in the target object appearance, and can generalize to novel objects with no prior exposure. Furthermore, the model provides a first-order approximation to predict human behavior during visual search.

## Results

**Visual search experiments**. We considered the problem of localizing a target object that could appear at any location in a cluttered scene under a variety of shapes, scales, rotations, and other transformations. We conducted 3 increasingly more difficult visual search experiments where 45 subjects had to move their eyes to find the target (Fig. 1, Methods). We propose a biologically inspired computational model to account for the fixations during visual search (Fig. 2).

**Searching for a target within an array of objects**. Many visual search studies have focused on images with isolated objects presented on a uniform background such as the ones in Experiment 1 (Figs. 1a and 3a). We used segmented grayscale objects from 6 categories from the MSCOCO dataset[40] (Methods). After fixation, 15 subjects were presented with an image containing a word describing the object category and a target object cue at a random 2D rotation (Fig. 1a). After an additional fixation delay, a search image was introduced, containing a different rendering of the target object, randomly located in one of 6 positions within a circle, along with 5 distractors from the other categories. The target was always present and appeared only once. The rendering of the target in the search image was different from the one in the target cue (e.g., Fig. 3a): it was a different exemplar from the same category, and it was shown at a different random 2D rotation. Subjects were instructed to rapidly move their eyes to find the target. Example fixation sequences from 5 subjects are shown in Fig. 3c: in these examples, subjects found the target in 1–4 fixations, despite the fact that the rendering of the target in the search image involved a different sheep, shown at a different 2D rotation. The target locations were uniformly distributed over the 6 possible positions (Supplementary Figure 1A) and subjects did not show any appreciable location biases (Supplementary Figure 1B). Subjects made their first fixation at 287 ± 152 ms (mean ± SD, $n = 15$ subjects, Fig. 3d). The interval between fixations was 338 ± 203 ms (Supplementary Figure 2A). The rapid deployment of eye movements is consistent with previous studies[10], and shows that subjects followed the instructions, without adopting alternative strategies such as holding fixation in the center and searching for the target purely via covert attention (Discussion).

Subjects located the target in 2.60 ± 0.22 fixations (mean ± SD, Fig. 3e), corresponding to 640 ± 498 ms (mean ± SD, Supplementary Figure 2B). The number of fixations required to find the

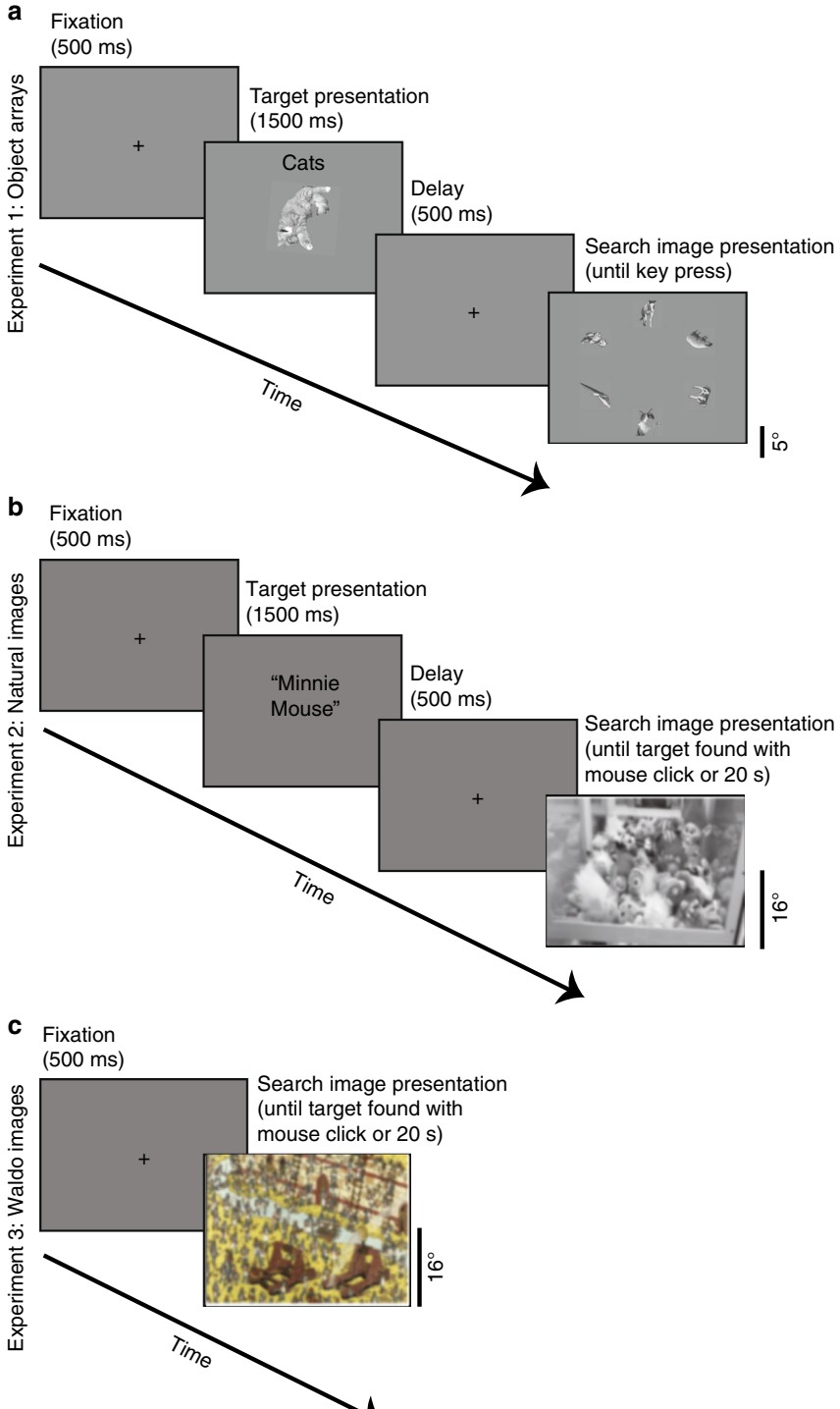

**Fig. 1** Schematic description of the three tasks. **a** Experiment 1 (Object arrays). **b** Experiment 2 (Natural images). **c** Experiment 3 (Waldo images). All tasks started with a 500 ms fixation period. Experiments 1 and 2 were followed by presentation of the target object for 1500 ms. In Experiment 1, the target object appeared at a random 2D rotation and the category descriptor was also shown to emphasize that subjects had to invariantly search for a different exemplar of the corresponding category shown at a different rotation. In Experiment 2, the target object was also different from the rendering in the search image. The target object (Waldo) was not shown in every trial in Experiment 3. In Experiments 1 and 2, there was an additional 500 ms delay after the target object presentation. Finally, the search image was presented and subjects had to move their eyes until they found the target. In Experiments 2 and 3, subjects also had to use the computer mouse to click on the target location. Due to copyright, in this and subsequent figures, all the search images were distorted by blurring. Due to copyright, we replaced the picture of a cat in **a** used in the experiment with a similar picture of a cat. Due to copyright, in **b**, during target presentation we replaced the picture used in the actual experiment with the text "Minnie Mouse". As noted under "Data availability", we made all the data including original images publicly available for research use

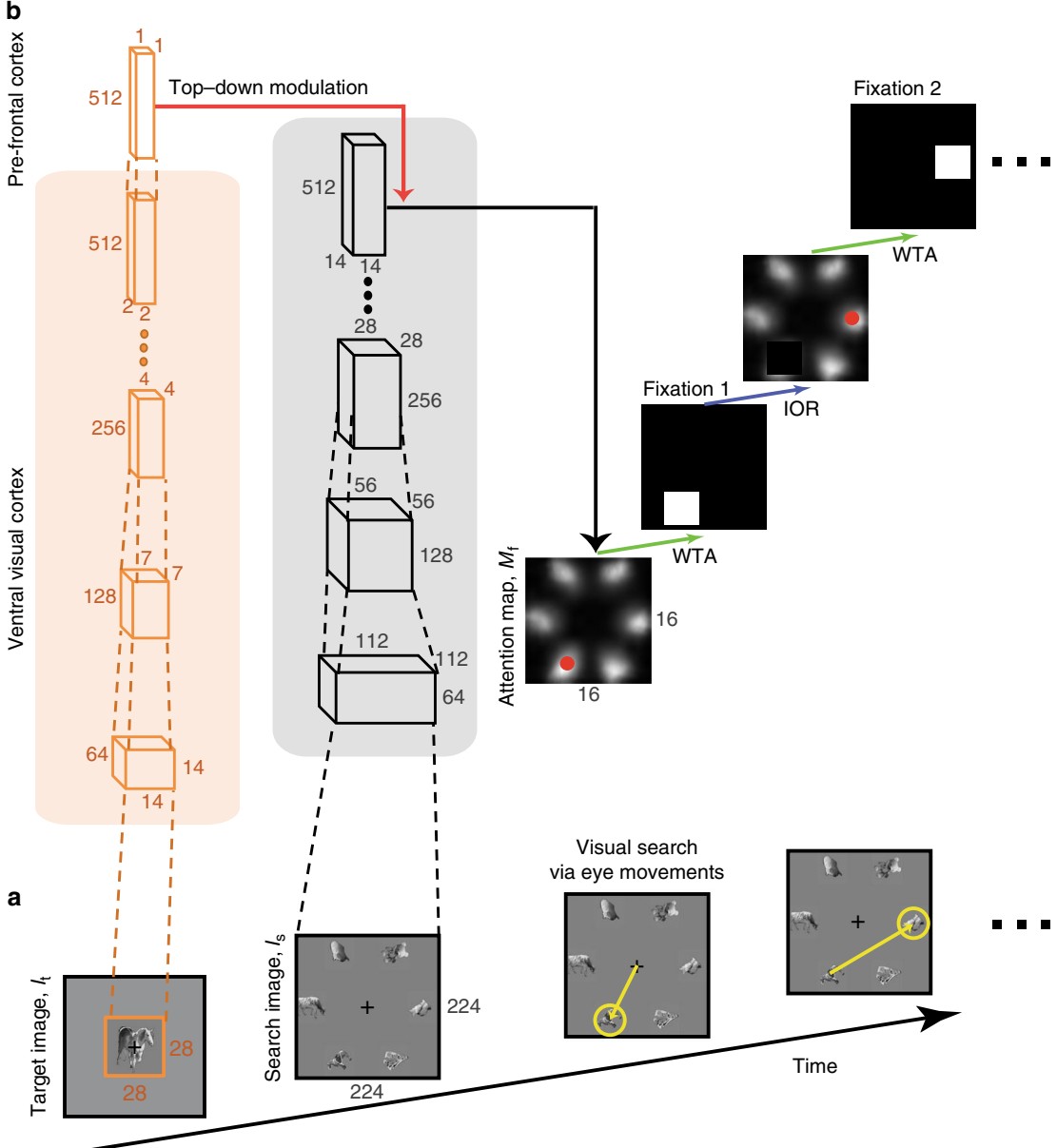

**Fig. 2** Model schematic. **a** Sequence of events during the visual search task. A target image is presented, followed by a search image where subjects move their eyes to locate the target object (see Fig. 1 for further details). **b** Architecture of the model, referred to as Invariant Visual Search Network (IVSN). The model consists of a pre-trained bottom-up hierarchical network (VGG-16) that mimics image processing in the ventral visual cortex for the target image (orange box) and for the search image (gray box). Only some of the layers are shown here for simplicity, the dimensions of the feature maps are indicated for each layer. The model generates features in each layer when presented with the target image $I_t$. The top-level features are stored in a pre-frontal cortex module that contains the task-dependent information about the target in each trial. Top-down information from pre-frontal cortex modulates (red arrow) the features obtained in response to the search image, $I_s$, by convolving the target presentation of $I_t$ with the top-level feature map from $I_s$, generating the attention map $M_f$. A winner-take-all mechanism (WTA, green arrow) chooses the maximum in the attention map (red dot) as the location for the next fixation. If the target is not found at the current fixation, inhibition of return is applied (IOR, blue arrow), the fixation location is set to 0 in the attention map and the next maximum is selected. This process is repeated until the target is found

target was significantly below the number expected from random exploration of the 6 possible locations, which would require 3.5 fixations in this experiment (Fig. 3e, $p < 10^{-15}$, two-tailed $t$-test, $t = 10$, df = 4473). Even in the first fixation, subjects were already better than expected by chance (performance = $26.4 \pm 4.1\%$ versus 16.7%). At 6 fixations, the cumulative performance was below 100% ($93.3 \pm 1.6\%$), since subjects revisited the same locations, even when they were wrong. The number of fixations required to find the target was lower when the target was identical in the target and search images (Supplementary Figure 3A-B), yet

subjects were able to efficiently and robustly locate the target despite changes in 2D rotation (Supplementary Figure 3B) and despite the exemplar differences (Supplementary Figure 3A).

To better understand the guidance mechanisms that incorporate target shape information to dictate the sequence of fixations, we implemented a computational model inspired by neurophysiological recordings in macaque monkeys during visual search tasks. The Invariant Visual Search Network (IVSN) model consists of a deep feed-forward network that mimics processing of features along ventral visual cortex, a way of temporally

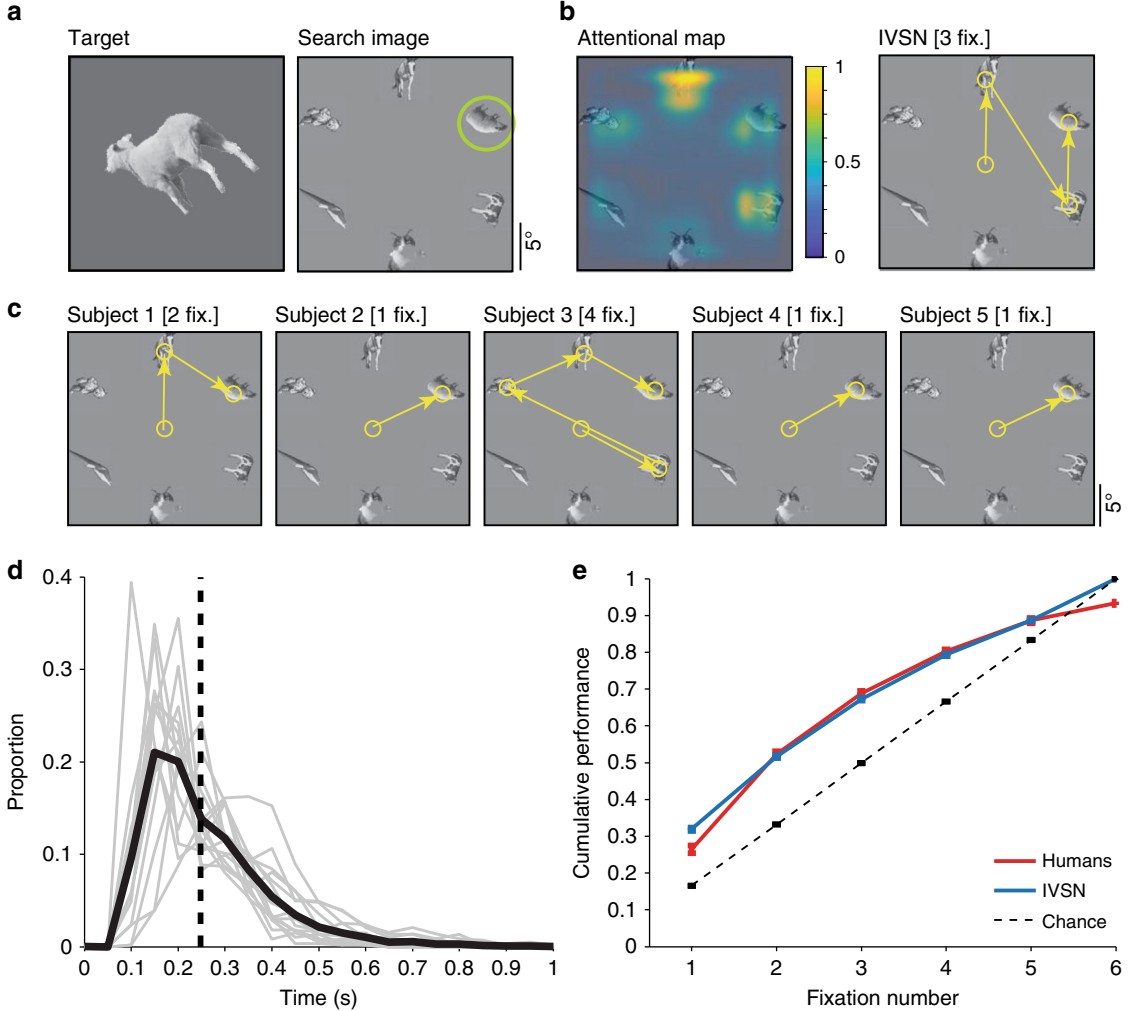

**Fig. 3** Experiment 1 (Object arrays). **a** Example target and search images for one trial. The green circle showing the target position was not shown during the actual experiment. Due to copyright, the picture of a sheep in the target image is not the actual one used in this experiment. **b** Attentional map overlaid on the search image and sequence of fixations for the IVSN model. **c** Sequence of fixations for 5 subjects (out of 15 subjects). The number above each subplot shows the number of fixations when the target was found. **d** Distribution of reaction times for the first fixation (see Supplementary Figure 2 for the corresponding distributions for subsequent fixations). Gray lines show the reaction time distributions for 5 individual subjects, the black line shows the average distribution. The median reaction time was 248 ms (SD = 130 ms, vertical dashed line). **e** Cumulative performance as a function of fixation number for humans (red), IVSN oracle model (blue), and chance model (dashed line). Error bars denote SEM, $n = 15$ subjects (see Supplementary Figure 4 for comparisons against other models)

storing information about the target tentatively associated with pre-frontal cortex, modulation of visual features in a top-down fashion to generate an attention map, and sequential selection of fixation locations (Fig. 2b, Methods). Of note, IVSN was neither trained with any of the images used in this study, nor was it trained in any way to match human performance. The same images used for the psychophysics experiments were presented to the model. The model builds an attention map (Fig. 3b, left) in response to the target and search images from Fig. 3a, and uses this map to generate a sequence of fixations, locating the target in 3 fixations (Fig. 3b, right). Despite the lack of training with this image set, and the large degree of heterogeneity between the target cue and the target's appearance in the search image, IVSN was able to efficiently locate the targets in $2.80 \pm 1.71$ fixations across all the trials (Fig. 3e, blue). IVSN performed well above the null chance model ($p < 10^{-11}$, two-tailed $t$-test, $t = 7.1$, df = 598), even in the first fixation (performance = $31.6 \pm 0.5\%$ compared to chance = $16.7\%$). The model had infinite inhibition-of-return and therefore never revisited the same location, by construction thus

achieving 100% performance at 6 fixations (see Supplementary Figure 11 and Discussion). Although there were no free parameters tuned to match human behavior, IVSN's performance was similar to that of humans. The strong resemblance between IVSN and human performance shown in Fig. 3e should not be over interpreted: there was still a small difference between the two ($p = 0.03$, two-tailed $t$-test, $t = 2.2$, df = 4473); in addition, we will discuss below other differences between humans and the IVSN model. Similar to human behavior, the model required fewer fixations when the rotation of the target cue matched the one in the search image, but the model was also able to efficiently locate the target at all the rotations tested (Supplementary Figure 3A-B).

We considered several alternative null models to further understand the image features that guide visual search (Supplementary Figure 4A). In the sliding window model, commonly used in computer vision, a fixed-size window sequentially scans the image (here scanning was restricted to the 6 locations), which is equivalent to random search with infinite inhibition of return

in this case, and fails to explain human behavior. Visual search was not driven by pure bottom-up saliency features as represented by the Itti and Koch model[26]. The weight features in the ventral visual cortex part of the model are important to generate the shape-invariant target-dependent visual attention map, as demonstrated by two observations: (i) randomizing those weights led to chance performance (RanWeight model); (ii) template matching algorithms based on pixels, using rotated templates or not, which are poor at invariant visual object recognition, were insufficient to explain human search behavior (Template Matching model). In sum, both humans and IVSN significantly outperformed all the alternative null models.

**Searching for a target in natural scenes.** The object array images used in Experiment 1 lack critical components of real-world visual search. In natural scenes, there is no fixed type and number of distractors equidistantly arranged in a circle, the target object is not segmented nor is it generally present on a uniform background, and the appearance of the target object can vary along multiple dimensions that are not pre-specified. In Experiment 2, we directly tackled visual search in natural images (Fig. 4). The structure of the task was essentially the same as that in Experiment 1 (Fig. 1b) with the following differences: (i) search images involved natural images (e.g., Fig. 4a), (ii) objects and distractors were not restricted to 6 categories, (iii) the appearance of the target object in the target and search images could vary along multiple dimensions, (iv) a trial was ended if the target was not found within 20 s, and (v) to ensure that the target was correctly found, subjects had to use the computer mouse to indicate the target location (Methods). The target locations were randomly and uniformly distributed (Supplementary Figure 1D). Subjects made rapid fixation sequences throughout the entire search image, with certain biases such as a larger density of fixations in the center and a smaller density of fixations along the borders (Supplementary Figure 1E). Fig. 4c shows example sequences where subjects were able to rapidly find the target in 2–5 fixations despite the changes in target appearance and despite the large amount of image clutter. The first fixation occurred at $285 \pm 135$ ms (Fig. 4d), and the interval between fixations was $290 \pm 197$ ms (Supplementary Figure 2C). The last fixations became progressively closer to the target (Supplementary Figure 2H). Subjects found the target in $1867 \pm 2551$ ms (Supplementary Figure 2D), which was about three times as long as in Experiment 1 (Supplementary Figure 2B).

Subjects located the target in $6.2 \pm 0.7$ fixations (Fig. 4e, red). Performance saturated at ~15 fixations, well below 100%. In $16.4 \pm 5.9\%$ of the images, subjects were unable to find the target within 20 s, hence human performance was well below ceiling. Human performance was more efficient than the chance model ($p < 10^{-15}$, two-tailed $t$-test, $t = 14$, df = 3247). Subjects tended to revisit the same locations even though the target was not there. In part because of this behavior, the null chance model showed a higher cumulative performance after 20 fixations. The average number of fixations that humans required to find the target was below that expected from the null chance model. Even in the first fixation, subjects were better than expected by chance (performance = $18.3 \pm 3.8\%$ versus $7.0 \pm 0.2\%$). The target as rendered in the search image could be larger or smaller than the target cue. Intuitively, it could be expected that performance might monotonically increase with the target size in the search image. However, subjects performed slightly better when the size of the target in the search image was similar to the original size in the target cue. Subjects were still able to robustly find the target across large changes in size (Supplementary Figure 3D). In addition to size changes, the target's appearance in the search image was

generally different in many other ways, which we quantified by computing the normalized Euclidian distance between the target cue and the target in the search image. Subjects robustly found the target despite large changes in its appearance (Supplementary Figure 3C).

Next, we investigated the performance of IVSN in natural images. Importantly, we used exactly the same model described for Experiment 1, with no additional tuning or any free parameters adjusted for Experiment 2. IVSN generated the attention map and scanpath in Fig. 4b in response to the target and search images from Fig. 4a: the model located the target in 3 fixations even though it had never encountered this target or any similar target before, despite the large amount of clutter, and despite the visual appearance changes in the target. IVSN efficiently located the target in natural scenes, requiring $8.3 \pm 7.5$ fixations on average (Fig. 4e, blue). IVSN performed well above the null chance model ($p < 10^{-15}$, two-tailed $t$-test, $t = 8.5$, df = 478), even in the first fixation ($14 \pm 5\%$ versus $7.0 \pm 0.2\%$). IVSN had infinite inhibition-of-return, never revisiting the same location, and achieving 100% accuracy in about 45 fixations. Humans outperformed the model up to approximately fixation number 10, but the model performed better than humans thereafter. Consistent with human behavior, IVSN was also robust to large differences between the size of the target in the search image and target cue (Supplementary Figure 3D) and it was also robust to other changes in target object appearance (Supplementary Figure 3C).

As described in Experiment 1, we considered several alternative null models, all of which were found to show lower performance than humans and IVSN (Supplementary Figure 4B). A pure bottom-up saliency model was worse than chance levels, because it did not incorporate features relevant to the target and instead concentrated on regions of high contrast in the image that were not relevant to the task. Similarly, template matching models were also worse than chance because they generated attention maps that emphasized regions that showed high pixel-level similarity to the target without incorporating invariance and therefore failing to account for the transformations in the target object shape present in the search image.

**Searching for Waldo.** The IVSN model could find objects that it had never encountered before (see also Supplementary Discussion and Supplementary Figure 5). To further investigate invariant visual search for novel objects, we designed Experiment 3 to test IVSN with more extreme images that bear no resemblance to those used in Experiments 1 and 2, or to the images in the ImageNet data set. We considered the traditional "Where is Waldo" task[41] (Fig. 5), comprising colorful cluttered drawings with scene statistics that are very different from those in natural images. The structure of Experiment 3 was similar to that of Experiment 2, except that a picture of Waldo was only presented at the beginning of the experiment and not in every trial (Fig. 1c). The target locations were randomly and uniformly distributed (Supplementary Figure 1G). Subjects made fixations throughout the entire search image, with certain biases such as a higher density in the center and a smaller density of fixations along the borders (Supplementary Figure 1H). Subjects made rapid sequences of fixations (e.g., Fig. 5c), with the first fixation occurring at $264 \pm 112$ ms (Fig. 5d), and an interval between fixations of $278 \pm 214$ ms (Supplementary Figure 2E). On average, subjects progressively became closer to the target in their last fixations (Supplementary Figure 2I).

Searching for Waldo constitutes a difficult challenge for humans, as confirmed by our results. On average, subjects found the target in $21.1 \pm 3.1$ fixations corresponding to $6051 \pm 4962$ ms

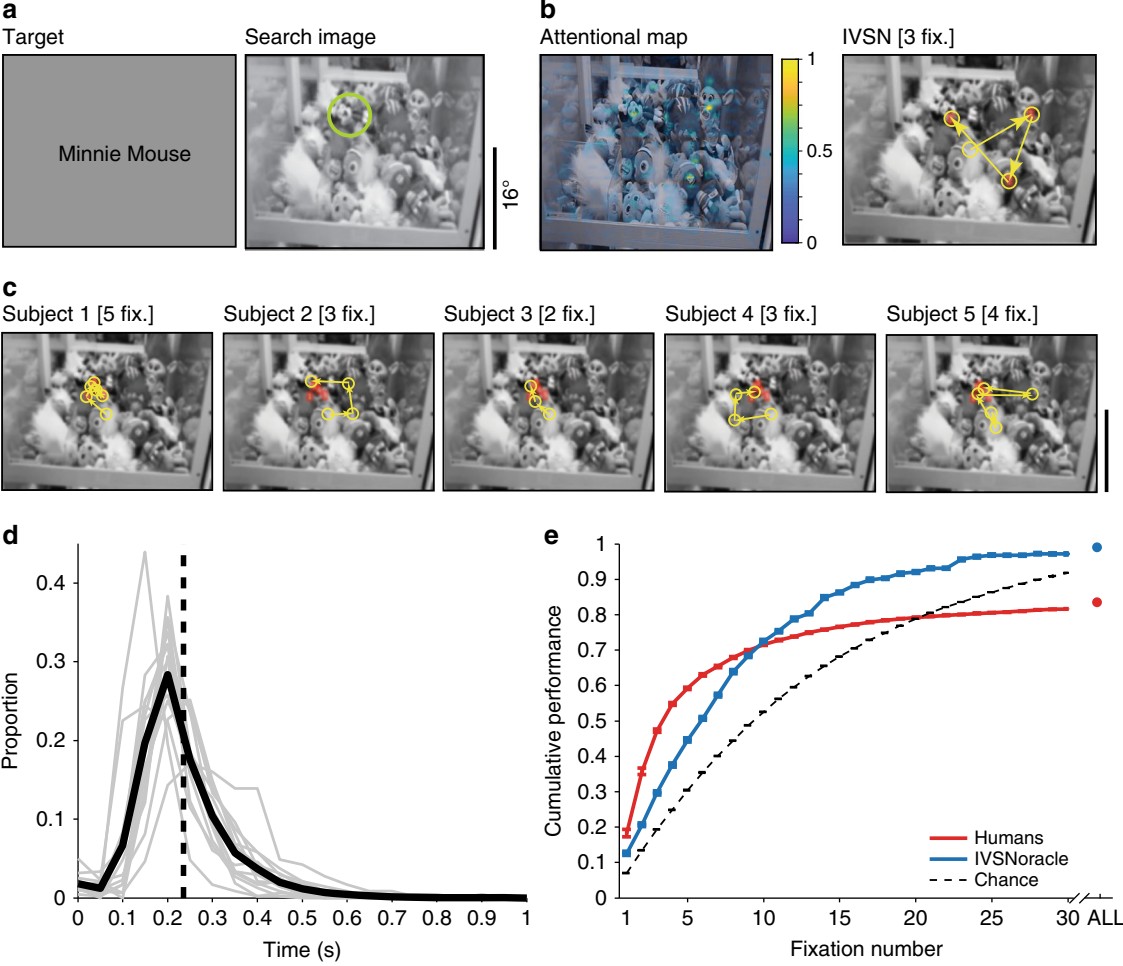

**Fig. 4** Experiment 2 (Natural images). The format and conventions follow those in Fig. 3. Due to copyright, we replaced the picture of Minnie Mouse shown during the actual experiment in the target image (**a**) with the text "Minnie Mouse"

(Fig. 5e, Supplementary Figure 2F), about three times longer than in Experiment 2 and about nine times longer than in Experiment 1. Performance reached a plateau at about 60 fixations, well below 100%. In 26.9 ± 9.6% of the images, subjects were unable to find the target within the allocated 20 s. Despite the task difficulty and despite infinite inhibition of return in the null chance model, subjects were able to find Waldo more efficiently than by random exploration ($p < 10^{-15}$, two-tailed $t$-test, $t = 18$, df $= 800$). There were also differences between the rendering of the target object in the search image and target image. Subjects were able to find Waldo despite these changes in target appearance (Supplementary Figure 3E).

We evaluated IVSN responses on the images from Experiment 3, without fine-tuning any parameters. IVSN had no prior experience with Waldo images or drawings of any kind. In the example in Fig. 5a, b, the model located Waldo in 9 fixations. IVSN efficiently located Waldo, requiring 29.0 ± 21.6 fixations on average (Fig. 5e, blue). IVSN performed well above the null chance model ($p < 10^{-15}$, two-tailed $t$-test, $t = 10$, df $= 116$). Despite the task difficulty, humans were more efficient in finding Waldo than IVSN ($p = 0.001$, two-tailed $t$-test, $t = 3.3$, df $= 784$). IVSN was robust to changes in the appearance of the target (Supplementary Figure 3E). The alternative null models did not perform as well as humans or the IVSN model (Supplementary Figure 4C).

Waldo was completely novel to IVSN but not for humans. We conducted a separate experiment with objects that were

completely novel for humans and showed that subjects were still able to find targets under situations where they had no prior exposure to the target objects (Supplementary Figure 10, Supplementary Discussion).

**Image-by-image comparisons**. The results presented thus far compared average performance between humans and models considering all images. We next examined consistency in the responses at the image-by-image level within-subjects (identical trials presented to the same subject), between-subjects, and between IVSN and subjects. We compared the number of fixations required to find the target in each trial in Supplementary Figure 7. Subjects were slightly more consistent with themselves than with other subjects, and the between-subject consistency was slightly higher than the consistency with IVSN (Supplementary Discussion).

The number of fixations provides a summary of the efficacy of visual search but does not capture the detailed spatiotemporal sequence of eye movements (Supplementary Figures 6, 8). We used the scanpath similarity score[27], to compare two fixation sequences (Supplementary Discussion). This metric captures the spatial and temporal distance between two saccade sequences, ranging from 0 (maximally different) to 1 (identical). Within-subject comparisons yielded slightly more similar sequences than between-subject comparisons in all 3 experiments (Fig. 6, $p < 10^{-9}$). The between-subject scanpath similarity scores,

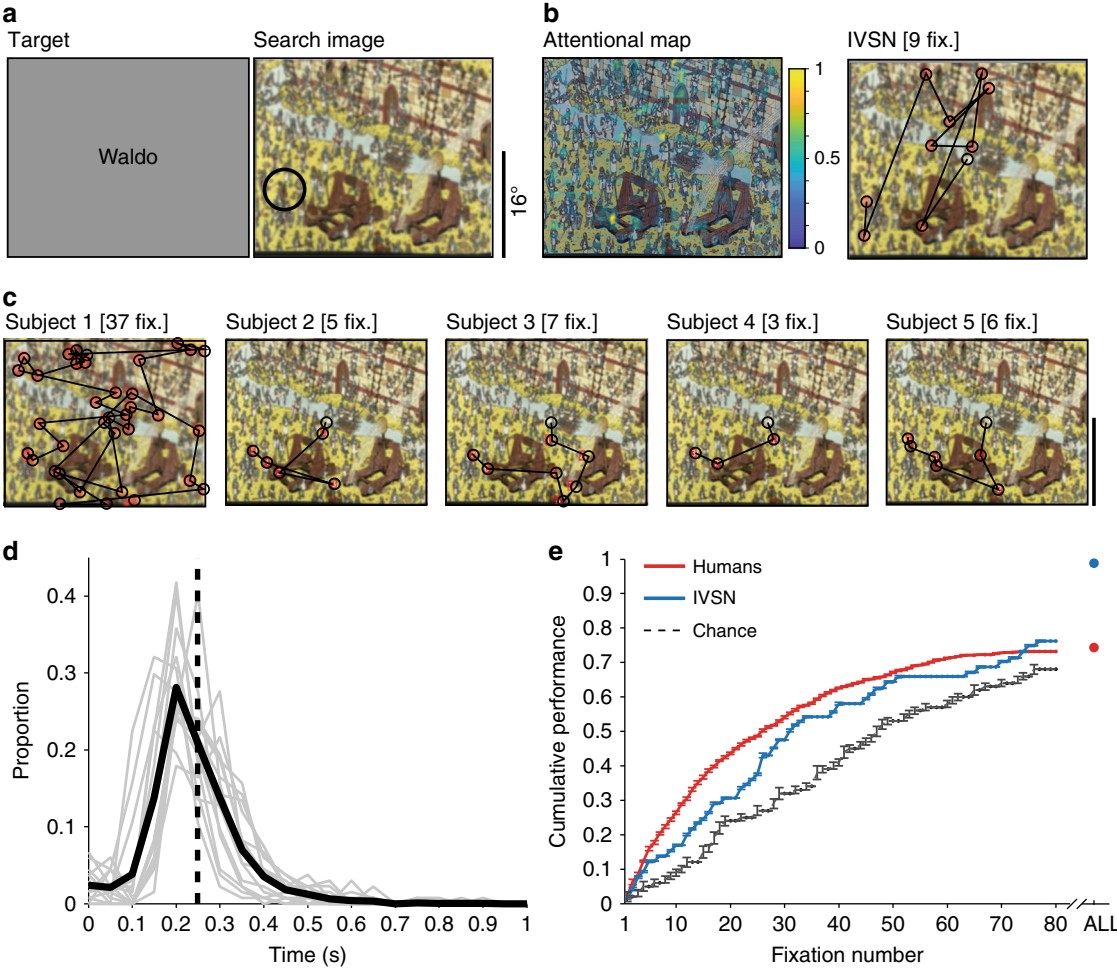

**Fig. 5** Experiment 3 (Waldo images). The format and conventions follow those in Fig. 3. Due to copyright, we replaced the picture of Waldo shown during the actual experiment in the target image (**a**) with the text "Waldo"

in turn, were higher than the IVSN-human similarity scores for all 3 experiments. The IVSN-human similarity scores were higher than the human-chance similarity scores for all 3 experiments. In sum, IVSN captured human eye movement behavior at the image-by-image level in terms of the number of fixations and the spatiotemporal pattern of fixations.

**Extensions and variations to the IVSN computational model.** We next considered variations of the IVSN model architecture and revisited several simplifications and assumptions of the model. The results presented thus far assumed that the model can perfectly recognize whether the target is present or not at the fixated location. After each fixation, an "oracle" decides whether the target is present or not. Rapidly recognizing whether the target is present or not is not easy, particularly in Experiments 2 and 3. Subjects sometimes fixated on the target, yet failed to recognize it, and continued the search process (Supplementary Figure 12A-B). Examples of this behavior are illustrated for Subjects 1 and 5 in Fig. 4c where the second fixations land on the target, yet the subjects make additional saccades and subsequently return to the target location. For fair comparison, all the psychophysics results presented thus far also used an oracle for recognition (search was deemed successful the first time that a fixation landed on the target). Without the oracle, human performance was lower but still well above chance (Experiment 2: $p$

$< 10^{-15}$, $t = 14$, df $= 3247$, Supplementary Figure 12C; Experiment 3: $p < 10^{-15}$, $t = 18$, df $= 800$, Supplementary Figure 12D). We introduced a simple recognition component into the model to detect whether the target was present or not based on the features of the object at the fixated location (IVSN$_{recognition}$, Supplementary Figure 11A-C, Methods). IVSN$_{recognition}$ performed slightly but not significantly below IVSN, particularly in the more challenging case of Experiment 2. IVSN$_{recognition}$ was still able to find the target above chance levels (Experiment 1: $p < 10^{-11}$, $t = 7.3$, df $= 594$, Supplementary Figure 11A; Experiment 2: $p < 10^{-13}$, $t = 8$, df $= 434$, Supplementary Figure 11B; Experiment 3: $p < 10^{-15}$, $t = 12$, df $= 112$, Supplementary Figure 11C).

Another simplification involved endowing IVSN with infinite inhibition of return. In contrast, humans show a finite memory and tend to revisit the same locations not only for the target (Supplementary Figure 12C-D) but also for non-target locations (e.g., subject 1 in Fig. 5c)[42]. We fitted an empirical function to describe the probability that subjects would revisit a location at fixation $i$ given that they had visited the same location at fixation $j < i$[42]. We incorporated this empirical function into the IVSN model so that previous fixated locations could be probabilistically revisited, thus creating a model with finite inhibition of return (IVSN$_{fIOR}$, Methods, Supplementary Figure 11D-F). The IVSN$_{fIOR}$ model showed lower performance than the IVSN model but this difference was not significant or marginally significant (Experiment 1: $p = 0.11$; Experiment 2: $p = 0.02$;

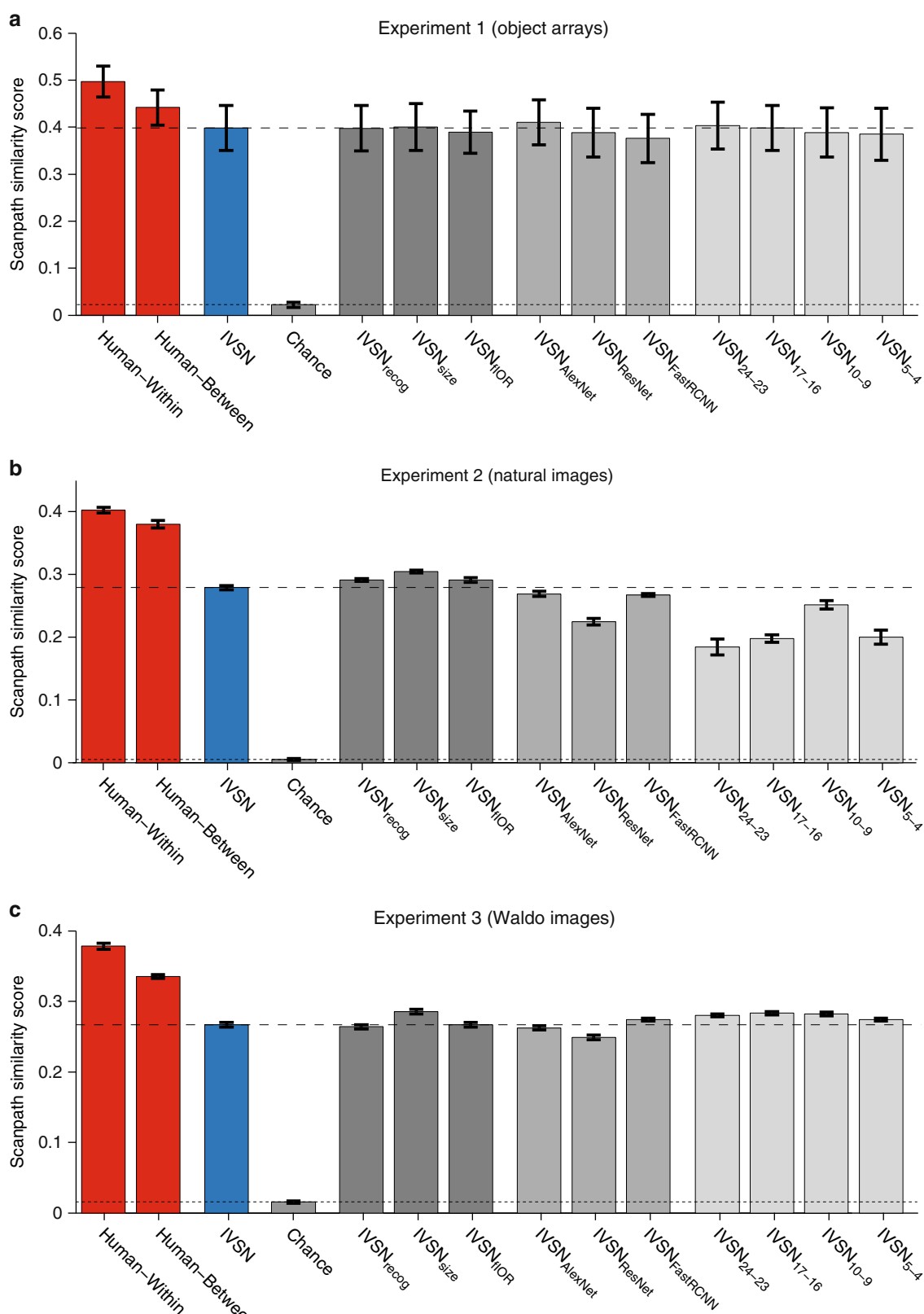

**a** Experiment 1 (object arrays)

**b** Experiment 2 (natural images)

**c** Experiment 3 (Waldo images)

Experiment 3: $p = 0.07$). Despite this drop in performance, IVSN$_{flOR}$ was still able to find the target better than chance (Experiment 1: $p = 10^{-15}$, $t = 9.7$, df $= 864$; Experiment 2: $p < 10^{-15}$, $t = 11$, df $= 617$; Experiment 3: $p = 10^{-15}$, $t = 16$, df $= 145$, two-tailed $t$-tests). Furthermore, IVSN$_{flOR}$'s performance was closer to humans for all 3 experiments (Supplementary Figure 11D-F, IVSN$_{flOR}$ versus human performance: Experiment 1: $p = 0.87$; Experiment 2: $p = 0.03$; Experiment 3: $p = 0.29$; two-tailed $t$-tests).

Another difference between humans and the model is the size of saccades (Supplementary Figure 11G-I). For example, in

**Fig. 6** Image-by-image consistency in the spatiotemporal pattern of fixation sequences. Scanpath similarity score (see text and Methods for definition) comparing the fixation sequences within subjects (first red column), between-subjects (second red column), between the default IVSN model and humans (blue column), and between all other models and humans (gray). Results are shown for Experiment 1 (**a**), Experiment 2 (**b**), and Experiment 3 (**c**). The larger the scanpath similarity score, the more similar the fixation sequences are. We considered sequences up to length 6, 30, and 80 in **a**, **b**, and **c**, respectively. Supplementary Figure 8A-C shows results comparing entire sequences and Supplementary Figure 8D-F shows results comparing scanpath similarity scores as a function of sequence length. The horizontal dashed line shows the IVSN-human similarity score and the dotted line shows the chance-human similarity score. Error bars denote SEM, $n = 15$ subjects. The within-subject similarity score was higher than the between-subject score in all 3 experiments ($p < 10^{-9}$). The between-subject similarity score was higher than the IVSN-human score in all 3 experiments ($p < 10^{-15}$) and the IVSN-human similarity scores were higher than human-chance scores in all 3 experiments ($p < 10^{-15}$)

Experiment 2, the average saccade size was $7.6 \pm 5.7°$ for humans and $16.8 \pm 8.4°$ for IVSN (Experiment 2: Supplementary Figure 11H, $p < 10^{-15}$, two-tailed $t$-test, $t = 62$, df $= 22,960$; Experiment 3: Supplementary Figure 11I, $p < 10^{-15}$, two-tailed $t$-test, $t = 100$ df $= 29,263$). Humans typically made relatively small saccades (Supplementary Figure 11H-I). In contrast, the saccade sizes for the model were approximately uniformly distributed (Supplementary Figure 11H-I). We used the empirical distribution of saccade sizes to probabilistically constrain the saccade sizes for the model, creating a new variation of the model, $IVSN_{size}$ (Methods) The distribution of saccade sizes for the $IVSN_{size}$ model resembled that of humans. $IVSN_{size}$ showed similar performance to IVSN (Experiment 1: $p = 0.97$; Experiment 2: $p = 0.52$; Experiment 3: $p = 0.47$; Supplementary Figure 11J-L), suggesting that the distribution of saccade sizes plays a lesser role in overall search efficiency.

Attentional modulation based on the target features is implemented in the IVSN model as a top-down signal from layer 31 to layer 30 in the VGG16 architecture (Fig. 2, Methods). Connectivity in cortex is characterized by ubiquitous top-down signals at *every* level of the ventral visual stream. We considered variations of the model where attention modulation was implemented via top-down signaling at different levels: layer 31 to 30 (default, Fig. 2), layer 24 to 23 ($IVSN_{24 \to 23}$), layer 17 to 16 ($IVSN_{17 \to 16}$), layer 10 to 9 ($IVSN_{10 \to 9}$), layer 5 to 4 ($IVSN_{5 \to 4}$) (Supplementary Figure 13). In general, these model variations were also able to find the target above chance levels (all models were statistically different from chance except for $IVSN_{5 \to 4}$ in Experiment 1). The low-level features (layer 5 to layer 4) showed the lowest performance, probably because they lack the degree of transformation invariance built along the ventral stream hierarchy. Generally, model features at higher levels showed better performance but the trend was not monotonic. For example, $IVSN_{24 \to 23}$ showed slightly better performance than IVSN in Experiment 1 (Supplementary Figure 13A), but this difference was not statistically significant ($p = 0.045$, two-tailed $t$-test, $t = 2$, df $= 299$).

We also considered the AlexNet[5], ResNet[43], and FastRCNN[21] architectures instead of the VGG16 architecture for the ventral visual cortex in Fig. 2 (Supplementary Figure 14). All of these alternative models were above chance in all the experiments ($p < 0.006$, Supplementary Discussion).

## Discussion
We examined 219,601 fixations to evaluate how humans search for a target object in a complex image under approximately realistic conditions and proposed a biologically plausible computational model that captures essential aspects of human visual search behavior. Subjects efficiently located the target in object arrays (Fig. 3), natural images (Fig. 4), and Waldo images (Fig. 5) despite large changes in the appearance of the target object when rendered in the search image. Search behavior could be approximated by a neurophysiology-inspired computational

network consisting of a bottom-up architecture resembling ventral visual cortex, a pre-frontal cortex-like mechanism to store the target information in working memory and provide top-down guidance for visual search, and a winner-take-all and inhibition-of-return mechanism to direct fixations. Both humans and the IVSN model, demonstrated selectivity, efficiency, and invariance, and did not require any training whatsoever with the sought targets.

Human visual search was efficient in that it required fewer fixations than alternative null models including random search, template matching, and sliding window models (Figs. 3e, 4e, and 5e). Humans actively sampled the image in a task-dependent manner, guiding search towards the target. Human visual search demonstrated invariance in being able to locate objects that were transformed between the target image and the search image in size (Experiments 1, 2, and 3), 2D rotation (Experiments 1 and 2), 3D rotation (Experiment 2), color (Experiment 3), different exemplars from the same category (Experiments 1 and 2), and other appearance changes including occlusion (Experiments 2 and 3). The large dissimilarity between how the targets were rendered in the search image and their appearance in the target image indicates that humans do not merely apply pixel-level template matching to find the target. These results suggest that the features guiding visual search must be invariant to target object transformations.

The problem of identifying objects invariantly to image transformations has been extensively discussed in the visual recognition literature (e.g., refs. [1,2,4], among many others). Indeed, the ventral visual cortex module in IVSN is taken from a computational model that is successful in object recognition tasks, VGG-16[3]. The invariance properties in IVSN are thus inherited from VGG-16. The current results show that the types of features learned upon training VGG-16 in an independent object labeling task (ImageNet[44]), can be useful not only in a bottom-up fashion for visual recognition, but also in a top-down fashion to guide feature-based attention changes during visual search. The current results show that top-down features guiding visual search must show invariance to object transformations.

There has been extensive work characterizing the features that guide visual search[9]. IVSN incorporates those ideas into a quantitative image-computable framework to explain how the brain decides where to allocate attention in a task-dependent manner. Importantly, there is no additional training in IVSN to achieve invariance. The current model, as well as other models of feature-based attention[10,11,29,33,45], assume that such top-down influences provide feature-selective and transformation-tolerant information. The lack of any training or fine-tuning in IVSN distinguishes the proposed model from other work in the object detection literature that focuses on supervised learning from a large battery of similar examples to locate a target[21,22]. The ability to perform a task without extensive supervised learning by extrapolating knowledge from one domain to a new domain is usually referred to as "zero-shot training". The specific exemplar objects in Experiments 1 and 2 were new to the subjects, even

though subjects had extensive experience with those object categories. Subjects were also able to efficiently search for novel objects from novel categories that they had never encountered before (Supplementary Figure 10). IVSN was able to find novel objects from known categories in Experiment 1. More strikingly, IVSN could find target objects in natural images even when those objects came from categories that it had never encountered before (Experiment 2, Supplementary Figure 5). Furthermore, IVSN could find Waldo in images that did not resemble any of the images used to train VGG-16 (Experiment 3). The ability to generalize and search for novel objects that have never been encountered before is consistent with the psychophysics literature showing that there are common feature attributes that guide visual search[9]. IVSN extends and formalizes the set of attributes from the low-level features that have been extensively studied in psychophysics experiments (e.g., color, orientation, etc.) to a richer and wider set of transformation-tolerant features relevant for visual recognition and for visual search under natural conditions.

Beyond exploring average overall performance, it is interesting to examine the spatiotemporal sequence of fixations for individual images. There is a large degree of variability when scrutinizing visual search at this high-resolution level. The same subject may follow a somewhat different eye movement trajectory when presented with the same exact target image and search image (Fig. 6, Supplementary Figures 7-8), an effect that cannot be accounted for by memory for the target locations (Supplementary Figure 7). As expected, the degree of self-consistency was higher than the degree of between-subject consistency, which was in turn higher than the degree of subject model consistency at the image-by-image level both for the number of fixations (Supplementary Figure 7) and for the spatiotemporal sequences of fixations (Fig. 6, Supplementary Figure 8).

Even when IVSN may approximate human search behavior, the model may not be searching in the same way that humans do. First, IVSN shows constant acuity over the entire visual field, which is clearly not the case for human vision where acuity drops rapidly from the fovea to the periphery. Second, humans must decide after each saccade whether the target is present or not. The default IVSN model executed this decision through an "oracle" (the same oracle was used for the human data for fair comparison, except in Supplementary Figure 12). As a proof-of-principle, we implemented a recognition step for each fixation in Supplementary Figure 11A-C, a step that can be improved through the extensive work on invariant visual recognition systems[1,3,5,43]. Humans also make recognition mistakes (e.g., Figs. 4c and 5c where subjects fixated on the target yet did not click the mouse, Supplementary Figure 12). Third, humans also revisit the same location even if the target is not there (e.g., Figs. 4c, 5c, 3e, 4e, and 5e[46,47]). Yet, the default IVSN model implements infinite inhibition of return as a simplifying assumption that could also be improved upon by including a memory decay function, as shown in IVSN$_{fIOR}$ (Supplementary Figure 11D-F). Fourth, there is no learning in the current model. The visual system could learn the interaction of the different bottom-up, top-down, memory and recognition components. An elegant idea on how learning could be implemented was presented in ref. [39] where the authors proposed an architecture that can learn to generate eye movements via reinforcement learning with a system that is rewarded when the target is found. IVSN can be improved by training or fine-tuning for various search tasks. Fifth, the model assumes that each saccade is independent of the previous one except for the inhibition-of-return mechanism and the saccade distance constraints. A complete model should incorporate inter-dependences across saccades such that visual information obtained during previous fixations can be used to guide the next saccade. Finally,

subjects may capitalize on high-level knowledge about scenes[9,48] including statistical correlations in object positions (e.g., car keys are usually not glued to the ceiling), physical properties (keys are more likely on top a desk rather than floating in the air), correlations in object sizes (the size of a phone may set an expectation for the size of the keys), etc.

As emphasized in the previous paragraph, there are multiple directions to improve our quantitative understanding of how humans actively explore a natural image during visual search. The current model provides a reasonable initial sketch that captures how humans can selectively localize a target object amongst distractors, the efficiency of visual search behavior, the critical ability to search for an object in an invariant manner, and zero-shot generalization to novel objects including the famous Waldo. Waldo cannot hide anymore.

## Methods

**Psychophysics experiments**. Participants: We conducted four psychophysics experiments with 60 naive observers (19–37 years old, 35 females, 15 subjects per experiment). The sample size was chosen based on the results in one of our previous experiments[10]. In Experiment 1, we used a sample size that was effective in a previous study with a similar structure[10]. For Experiments 2 and 3, we used the same sample size to facilitate comparisons across experiments. We focus on the first 3 experiments in the main text and report the results of the fourth experiment in Supplementary Figure 10. All participants had normal or corrected-to-normal vision. Participants provided written informed consent and received 15 USD per hour for participation in the experiments, which typically took an hour and a half to complete. All the psychophysics experiments were conducted with the subjects' informed consent and according to the protocols approved by the Institutional Review Board at Children's Hospital.

Experimental protocol: The general structure for all three experiments was similar (Fig. 1). Subjects had to fixate on a cross shown in the middle of the screen, a target object was presented followed by another fixation delay (Experiments 1 and 2), a search image was presented, and subjects had to move their eyes to find the target. In Experiments 2 and 3, subjects also had to indicate the target location via a mouse click. Stimulus presentation was controlled by custom code written in MATLAB using Version 3.0 of the Psychophysics Toolbox[49]. Images were presented on a 19-in. CRT monitor (Sony Multiscan G520), at a 1024 × 1280 pixel resolution, subtending approximately 32 × 40° of visual angle. Observers were seated at a viewing distance of approximately 52 cm. We recorded the participants' eye movements using the EyeLink D1000 system (SR Research, Canada).

Experiment 1 (Object arrays): We selected segmented objects without occlusion from 6 categories in the MSCOCO dataset of natural images[40]: sheep, cattle, cats, horses, teddy bears, and kites (e.g., Fig. 3a). Due to the uncontrolled and diverse nature of stimuli in the MSCOCO dataset, the images may differ in low-level properties that could contribute to visual search performance. To minimize such contributions, we took the following steps: (1) resized the object areas such that a bounding box of 156 × 156 pixels encompassed the outermost contour of the object while maintaining their aspect ratios; (2) converted the images to grayscale; (3) equalized their luminance histograms; and (4) randomly rotated the objects in 2D. We conducted a verification test to make sure that the low-level features of all the objects were minimally discriminative: we considered the feature maps from the first convolution blocks of four pre-trained image classification networks (ResNet[43], AlexNet[5], VGG16 and VGG19[3]), and performed cross-validated category classification tests on these features maps as well as on the image pixels using a Support Vector Machine (SVM) classifier[50]. The total of 2000 object images were split into 5 groups for training, validation, and testing. The classification performance obtained with these low-level features was consistent across the different computational models and was slightly above chance levels (Supplementary Table 1).

A schematic of the sequence of events during the task is shown in Fig. 1a. After fixation for 500 ms, a random exemplar from the target category was shown in the fixation location, subtending 5.5° of visual angle, for 1500 ms. The object was shown at a random rotation (0–360°) along with the category name. After another 500 ms of fixation, the search image was presented. Subjects searched for the target in a search image containing an array of 6 objects (Fig. 3a). In the search images, the 6 objects, each 156 × 156 pixels and subtending ~5° of visual angle, were uniformly distributed on a circle with a radius of 10.5° eccentricity. All the objects could be readily recognized by humans at this size and eccentricity. The target was always present only once within these 6 objects and was placed randomly in one of the 6 possible positions. Supplementary Figure 1A shows the distribution of target object locations. There was one distractor from each category, randomly chosen.

Subjects were instructed to find the target as soon as possible by moving their eyes and pressed a key to go to the next trial. To evaluate within-subject consistency, and unbeknown to the subjects, each trial was shown twice (the exact same target image and search image were repeated). The order of all trials was randomized. There were 300 × 2 = 600 trials in total, divided into 10 blocks of 60

trials each. We split the 300 unique trials into 180 target-different trials and 120 target-identical trials (Supplementary Figure 9A). In the target-identical trials, the appearance of the target object within the search image was identical to that in the target image. In the target-different trials, the target object was a random exemplar from the same category as the one shown in the target image, and was presented at a random rotation (0–360°). Target-different and target-identical trials were randomly interleaved, except in the additional experiment discussed in Supplementary Figure 9D (see below). To evaluate between-subject consistency, the same target and search images were shown to different subjects.

We initially hypothesized that performance would be higher in target-identical trials compared to target-different trials. Upon examining the results, this hypothesis was found to be correct but the difference in performance between target-identical and target-different trials was small (Supplementary Figure 9C). In addition, performance in the target-identical trials was lower than what we reported previously in a different experiment consisting exclusively of target-identical trials and using different objects[10]. We conjectured that the task instructions and structure including the presence of target-different trials influenced performance in the target-identical trials. To further investigate this possibility, we conducted an additional variation of Experiment 1 in which target-identical and target-different trials were blocked (Supplementary Figure 9D). In this task variation, subjects were told whether the next block would include target-identical or target-different trials. To counter-balance any presentation order biases, we tested 2 subjects on target-identical trials first followed by target-different trials and 3 subjects on the reversed order. This experiment confirmed our intuitions and showed that performance was higher in target-identical trials when they were blocked, compared to when they were interleaved, while performance in target-different trials did not depend on the task structure and instructions. Throughout the text (and except for Supplementary Figure 9D), we focus all the analyses on the original and more natural version of the task where target-identical and target-different trials were randomly interleaved.

Experiment 2 (Natural images): We considered 240 objects from common object categories, such as animals (e.g., clownfish) and daily objects (e.g., alarm clock). The object sizes were 106.5 ± 71.9 pixels high × 114.4 ± 74.8 pixels wide. The 240 objects were *not* restricted to the 6 categories in Experiment 1 but could involve any object. To test whether IVSN can generalize to searching for novel objects (zero-shot training), we also included objects that are *not* part of the 2012 ImageNet data set[44] (the database of images used to train the model, see Model section below). Examples of such objects include SpongeBob toys, Eve robot, Ironman figures, QuickTime app icon, deformed flags or clothes, weapons, tamarind fruits, fried chicken wings, special hand gesture, Lego blocks, push toys, chopsticks, and ribbons on gifts, among others. There were 140 images out of the selected 240 images containing target objects that were not included in ImageNet. All target objects were manually selected such that each search image contained only one target object. The object shown in the target image was *not* segmented from the search image, but rather was a similar object: for example, Fig. 4a shows a vertically and rotated version of "Minnie" with a dress and bow displaying white circles (left) whereas the target is rendered in the search image shows Minnie at a different scale, with a different attire, partially occluded and under different rotation (right). The search images were 1028 × 1280 pixel natural images that contained the target amidst multiple distractors and clutter (e.g., Fig. 4a). Both the search images and the target images were presented in grayscale. As illustrated in Fig. 4a, the target objects were picked such that they were visually different from the ones rendered on the search images; these changes included changes in scale, 2D and 3D rotation, changes in attire, partial occlusion, etc.

The sequence of steps in Experiment 2 followed the one described for Experiment 1 (Fig. 1b), with three differences described next. The presentation of the target image did not include any text. The search image was a grayscale natural image, always containing the target, and occupied the full monitor screen (subtending ~32 × 40° of visual angle). Supplementary Figure 1B shows the distribution of target object sizes and locations within the search image, which were approximately uniformly distributed. The appearance of the target object within the search array was always different from that in the target image, that is, there were no target-identical trials. Subjects were instructed to find the target as soon as possible by moving their eyes. Experiment 2 was harder than Experiment 1 because objects in the search image were not segmented and were shown embedded in complex natural clutter, and because the appearance of the target object was more different from the target object than in Experiment 1 (e.g., compare *x*-axis in Supplementary Figure 3A versus S3C versus S3E). As the search task became more difficult, subjects would fixate on the target object, yet fail to realize that they had landed on the target (Supplementary Figure 12). Hence, to ensure that subjects had consciously found the target, they had to use the computer mouse to click on the target location. If the clicked location fell within the ground truth, subjects went on to the next trial; otherwise, subjects stayed on the same search image until the target was found. If the subjects could not find the target within 20 s, the trial was aborted, and the next trial was presented. Subjects were unable to find the target within 20 s in 16.4% of the trials. To evaluate between-subject consistency, different subjects were presented with the same images. To evaluate within-subject consistency, every trial was repeated once, in random order (same target image and same search image). To avoid any potential memory effect (whereby subjects could remember the location of the target), we restricted the analyses to the first presentation, except in the within-subject consistency metrics reported in

Supplementary Figures 6, 7, and 8. The results were very similar for the first instance of each image versus the second instance of each image and any memory effects across trials were minimal, but we still implemented these precautions focusing the results on the first instance of each image in all the experiments.

Experiment 3 (Waldo images): "Where's Waldo" is a well-known search task[41] with crowded scene drawings containing hundreds of individuals that look similar to Waldo undertaking various activities. Exactly one of these individuals is the character known as Waldo (e.g., Fig. 5a). We tested 67 Waldo images from ref. [41]. The target object sizes were 24.7 ± 4.5 pixels wide and 40.3 × 7.4 pixels high. Given the large size of the Waldo search images and the limited precision of our eye tracker in terms of individual characters on these images, we cropped each Waldo image into four quadrants and only showed the human subjects the quadrant containing Waldo. There were 13 out of 67 images that had an instruction panel in the upper left corner that could contain additional renderings of Waldo. Subjects were explicitly instructed not to look at the instruction panel. At the model evaluation stage, these areas were also discarded. The locations of these panels can be approximately glimpsed from less dense fixation patches in Supplementary Figure 1H. Because all subjects were familiar with the Waldo task, we changed the overall structure such that there was no target image presentation in each trial (Fig. 1c). The target (Waldo) in color was presented at the beginning of the experiment. After fixation, the search image, always containing Waldo, was presented occupying the full monitor screen (subtending ~32 × 40° of visual angle). Subjects were instructed to find Waldo as soon as possible by moving their eyes. Similar to Experiment 2, once the target was found, subjects had to click on the target location. If the clicked location fell on the ground truth, subjects proceeded to the next trial; otherwise, subjects stayed on the same search image until the target was found. If subjects could not find the target in 20 s, the trial was aborted. The limit of 20 s was based on pilot tests and was dictated by a compromise between allowing enough time to find the target in as many trials as possible while at the same time maximizing the number of search trials. Subjects were unable to find the target within 20 s in 27% of the trials. There were 67 trials in total and the trial order was randomized. Within- and between-subject consistency was evaluated as described above for Experiments 1 and 2. In addition to searching for Waldo, we conducted a separate set of trials where subjects searched for the "Wizard", another character in the Waldo series. The results for the Wizard search were similar to those for the Waldo search. We restrict this report to the Waldo search task for simplicity.

Experiment 4 (Novel objects): We conducted an additional experiment to evaluate whether human subjects are able to search for novel objects that they have never encountered before (other than the single exposure to the target image). We collected a total of 1860 novel objects belonging to 98 categories. These objects were composed from well-designed novel object parts and we also included novel objects used in previous studies (Supplementary Figure 10)[51,52]. We used the same pre-processing steps to normalize the novel objects' low-level features as in Experiment 1. Supplementary Figure 10A shows 6 example novel objects. The task structure followed the one in Experiment 1, except that here there was no text indicating the object category during the target presentation (Supplementary Figure 10B). The number of trials for target identical and target different trials was balanced (80 target-identical versus 80 target-different trials in novel objects). To directly compare the results for novel objects versus those obtained with known objects, the objects from Experiment 1 (known objects) were also presented in this experiment, randomly intermixed with the novel object trials.

In visual search experiments, the similarity between the target object and the distractor objects plays a critical role in the difficulty of the task. As a proxy for task difficulty, we computed the similarity between the target object and the distractors by computing the Euclidian distance between all possible target–distractor object pairs in each image (*x*-axis in Supplementary Figure 10C). The target and distractor novel objects were chosen so as to match the distribution of similarities for known objects (Supplementary Figure 10C) to avoid scenarios where one set of stimuli could be easier to discriminate than in the other set. The results for the novel object visual search experiment are shown in Supplementary Figures 10D-E.

**Visual search computational models**. We first provide a high-level intuitive outline of our IVSN model, followed by a full description of the implementation details. IVSN posits an attention map, $M_f$, which determines the fixation location by conjugating local visual inputs with target information (Fig. 2). Both the target image ($I_t$) and the search image ($I_s$) are processed through the same deep convolutional neural network, which aims to mimic the transformation of pixel-like inputs through the ventral visual cortex[1,2,4]. Feature information from the top level of the visual hierarchy is stored in a module, which we refer to as pre-frontal cortex, based on the neurophysiological role of this area during visual search (e.g., ref. [15]). Activity from the pre-frontal cortex module provides top-down modulation, based on the target high-level features, on the responses to the search image, generating the attention map $M_f$. A winner-take-all mechanism selects the maximum local activity in the attention map $M_f$ for the next fixation. If the fixation location contains the target, the search stops. Otherwise, an inhibition-of-return mechanism leads the model to select the next maximum in the attention map and the process thus continues until the target object is found. The model was always presented with the exact same images that were shown to the subjects in the psychophysics experiments described in the previous section.

Ventral visual cortex: The deep feed-forward network builds upon the basic bottom-up architecture for visual recognition described in previous studies (e.g., [1–8]). We used a state-of-the-art deep feed-forward network, implemented in VGG16[3], pre-trained for image classification on the 2012 version of the ImageNet dataset[44]. The network weights **W** learnt from image classification extract feature maps for an input image of size $224 \times 224$ pixels. The same set of weights, that is, the same network, is used to process the target image and the search image. Only a subset of the multiple layers is illustrated in Fig. 2 for simplicity (see ref. [3] for full details of the VGG16 architecture). The images from the ImageNet dataset used to train the ventral visual cortex network for object classification are different from all the images used in the experiments. The 6 categories from MSCOCO in Experiment 1 are also present in ImageNet. In Experiment 2, 140 of the 240 target objects were not part of the 1000 ImageNet categories. None of the images in Experiment 3 or in the novel object experiment (Supplementary Figure 10) had any resemblance to the categories in ImageNet. The weights **W** do not depend on any of the target images $I_t$ or the search images $I_s$ (hence the model constitutes a zero-shot training architecture for visual search). The output of the ventral visual cortex module is given by the activations at the top-level (Layer 31 in VGG16[3]), $\varphi_{31}$ ($I_t$, **W**), and the layer before that (Layer 30 in VGG16), $\varphi_{30}$ ($I_s$, **W**), in response to the target image and search image, respectively (in Supplementary Figure 13 we considered top-down modulation between different layers). As noted above, it is the same exact network, with the same weights **W** that processes the target and search images, and we use the activations in layer 31 in response to the target image to provide top-down modulation to layer 30's response to the search image (Fig. 2). In Experiments 2 and 3, the images were too large ($1080 \times 1240$ pixels) for the model and down-sampling the images would make the finely detailed characters hard to discern. Therefore, we partitioned the whole image into segments of size $224 \times 224$, repeatedly ran the model in each of these segments and finally concatenated the resulting attention maps.

Pre-frontal cortex: The top-level of the VGG-16 architecture conveys the target image information to the pre-frontal cortex module, consisting of a vector of size 512. To search for the target object, IVSN uses the ventral visual cortex responses to that target image stored in the pre-frontal cortex to modulate the ventral visual cortex responses to the search image. This modulation is achieved by convolving the representation of the target with the representation of the search image before max-pooling:

$$M_f = m(\varphi(I_t, \mathbf{W}), \varphi(I_s, \mathbf{W})) = m(\varphi_{31}(I_t, \mathbf{W}), \varphi_{30}(I_s, \mathbf{W}))$$

where $m(.)$ is the target modulation function defined as a 2D convolution operation with kernel $\varphi_{31}$ ($I_t$, **W**) on the search feature map $\varphi_{30}$ ($I_s$, **W**). $M_f$ denotes the attention map.

Fixation sequence generation: At any point, the maximum in the attention map determines the location of the next fixation. In the figures, we normalize the attention map to [0,1] for visualization purposes.

A winner-take-all mechanism selects the fixation location. The model needs to decide whether the target is present at the selected location or not (see below). If the target is located, search ends. Otherwise, inhibition-of-return[47] is applied to $M_f$ by reducing the activation to zero in an area of pre-defined size ($45 \times 45$ pixels in Experiment 1, $200 \times 200$ in Experiment 2, $100 \times 100$ in Experiment 3), centered on the current fixation location. This reduction is permanent, in other words, infinite memory is assumed for inhibition of return here. These window size choices were based on the average object sizes in each experiment. Similar to other attention models (e.g., ref. [26]), the winner-take-all mechanism then selects the next fixation location and this procedure is iterated until the target is found. In the psychophysics experiments, we limited the duration of each trial to 20 s. When we compared the number of fixations at the image-by-image level (Supplementary Figure 7), we restricted the analyses to those images when the target was found and excluded those images where the target was not found in 20 s (see previous section for percentages in each task). Otherwise, all images were included in the analyses.

Target presence decision: Given a fixation location, the model needs to perform visual recognition to decide whether the target is present or not (in a similar way that humans need to decide whether they found the target after moving their eyes to a new location). There has been extensive work on visual recognition models (e.g., refs. [1,3–5]). In this study, we focus on the attention selection mechanism. To isolate the search process from the verification process, in the default IVSN model we bypass the recognition question by using an "oracle" system that decides whether the target is present or not (see Supplementary Figure 11A-C for IVSN$_{recognition}$). The oracle checks whether the selected fixation falls within the ground truth location, defined as the bounding box of the target object. The bounding box is defined as the smallest square encompassing all pixels of the object. For fair comparison between models and humans, we implemented the same oracle system for the human psychophysics data (except in Supplementary Figure 11D-F, 12), by considering the target to be found the first time a subject fixated on it.

Comparison with other models: We performed several comparisons with other models (Supplementary Figures 4, 11, 13, 14). In all cases, the alternative models proposed a series of fixations. In all cases except for IVSN$_{recognition}$ (described below), we used the oracle method to decide whether to stop search or to move on to the next fixation. In all cases except for IVSN$_{fIOR}$ (described below), the models

had infinite inhibition of return (IOR), as described above. We considered the following alternative models:

(1) Chance. We considered a model where the location of each fixation was chosen at random. In Experiment 1, we randomly chose one out of the six possible locations, while still respecting infinite IOR. In Experiments 2 and 3, a random location was selected in each fixation, while still respecting IOR; this random process was repeated 100 times. The selected location was the center of a window of the same size used for the recognition model described above. This window was used to determine the presence of the target and also to set IOR.

(2) Sliding Window (SW). We considered a sliding window approach which takes the fixated area (a window of the same size used for the recognition model described above) as inputs, scans the search image from the top left corner with stride 28 pixels, and uses oracle verification to determine target presence. In Experiment 1, the sliding window sequentially moves through the 6 possible objects.

(3) Template Matching. To evaluate whether pixel-level features of the target were sufficient to direct attention, we introduced a pixel-level template-matching model where the attention map was generated by sliding the canonical target of size $28 \times 28$ pixels over the whole search image. Compared with the SW model, the Template Matching model can be thought of as an attention sliding window.

(4) IttiKoch. It is conceivable that in some cases, attention selection could be purely driven by bottom-up saliency effects rather than target-specific top-down attention modulation. We considered a pure bottom-up saliency model that has no information about the target[26].

(5) RanWeight. Instead of using VGG16[3], pre-trained for image classification, we randomly picked weights **W** from a Gaussian distribution with mean 0 and standard deviation 1000. The network was otherwise identical to IVSN. We ran 30 iterations of this model, each iteration with random selection of weights.

Variations and extensions of the IVSN model: We considered several possible extensions and variations of the IVSN model.

IVSN$_{AlexNet}$ (Supplementary Figure 14). The "ventral visual cortex" module in Fig. 2 was replaced by the AlexNet architecture[5]. The "pre-frontal cortex" module corresponded to layer 8 and sent top-down signals to layer 7.

IVSN$_{ResNet}$ (Supplementary Figure 14). The "ventral visual cortex" module in Fig. 2 was replaced by the ResNet200 architecture[43]. The "pre-frontal cortex" module corresponded to the output of residual block 8 in the target image and sent top-down signals to residual block 8 in the search image.

IVSN$_{FastRCNN}$ (Supplementary Figure 14). The "ventral visual cortex" module in Fig. 2 was replaced by the FastRCNN architecture[21] pre-trained on ImageNet for region proposal and pre-trained on PASCAL VOC for object detection. The "pre-frontal cortex" module corresponded to layer 24 and sent top-down signals to layer 23.

IVSN$_{24\to23}$, IVSN$_{17\to16}$, IVSN$_{10\to9}$, IVSN$_{5\to4}$ (Supplementary Figure 13). In the IVSN model as presented in Fig. 2 (based on the VGG16 architecture[3]), the "pre-frontal cortex" module corresponded to layer 31 and sent top-down signals to layer 30. We considered several variations using top-down features from different levels of the VGG16 architecture as described by the model sub-indices.

IVSN$_{recognition}$ (Supplementary Figure 11A-C). The IVSN model presented in the main text uses an oracle to determine whether the target was found at a given fixation or not. In the brain, of course, there is no oracle. Each fixation places the new location within the high-resolution fovea, and responses along the ventral visual stream within this region are enhanced via attention modulation[15,29,31]. By emphasizing the selected areas, IVSN allows the ventral pathway to perform fine-grained object recognition. As a schematic proof-of-principle of a model that addresses whether the target was found or not, in Supplementary Figure 11A-C we implemented an additional step that included recognition after fixation. This recognition machinery involved an object classifier which determined whether the fixated area contained the target or not (IVSN$_{recognition}$). We implemented this step by cropping the search image centered at the fixation location using the same window sizes described for inhibition of return ($45 \times 45$, $200 \times 200$, and $100 \times 100$, for Experiments 1, 2, and 3, respectively), and using the object recognition network, VGG16[3], pre-trained on ImageNet[44], to extract the classification vector from the last layer, which emulates responses in inferior temporal cortex with high object selectivity and large receptive fields, for both the target image $I_t$ and the cropped area. The Euclidean distance between activation of this top layer to $I_t$ and the cropped area was computed. If this Euclidian distance was below a threshold of 0.9, the target was deemed to be found and search was stopped. Otherwise, the search continued after applying inhibition-of-return, as described above for the oracle. In this model including a recognition component, failure to locate the target could be due to fixating on the wrong location or fixating on the right location but not realizing that the target was there.

IVSN$_{fIOR}$. The IVSN model assumes infinite inhibition-of-return, that is the model never revisits a given fixation location. In contrast, humans do tend to revisit the same location even if the target is not there. An example of this behavior can be seen in multiple fixations from subject 1 in Supplementary Figure 5C and also in

fixations 3 and 6 in Supplementary Figure 7B2 (the reader may have to zoom in on the figures to appreciate this phenomenon). The finite inhibition of return is a well-known phenomenon in the psychophysics literature[42,46,47]. We implemented a variation of the IVSN model with finite inhibition-of-return (IVSN$_{fIOR}$). At each location in the image $(x,y)$ and at time $t$, the feature attention map $M_f$ was multiplied by a memory function $M_m$ to generate a new attention map $A_f(x,y) = M_f(x,y){*}M_m(x,y,t)$. In the implementation with infinite IOR, $M_m(x,y,t)$ is 0 if the location $(x,y)$ was visited previously and 1 otherwise (independently of time $t$). In the IVSN$_{fIOR}$ model, $M_m(x,y,t)$ was fitted to the empirical probability of revisiting a location from the human psychophysics data. The inaccuracy in our eye movement measurements is on the order of 1° of visual angle. To be overly cautious, we defined a location as revisited if another fixation landed within 3° of visual angle. None of the parameters in the default IVSN model were trained or fitted to human psychophysics data. In contrast, the function $M_m$ was fitted to the human psychophysics data, separately for each experiment. To avoid overfitting, we randomly selected 7 out of the 15 subjects to fit $M_m$ and all the comparisons between IVSN$_{fIOR}$ and human psychophysics were based on the remaining 8 subjects.

IVSN$_{size}$. The IVSN model has no constrain on the size of each saccade (e.g., one fixation could be in the upper left corner and the immediate next fixation could be in the lower right corner). In contrast, humans tend to make smaller saccades following a gamma-like distribution (Supplementary Figure 11G-I). We implemented a variation of the IVSN model where the saccade size was constrained by the empirical distribution of human saccade sizes (IVSN$_{size}$). We defined the attention map as a weighted sum of the feature attention map $M_f$ and a size constraint function $M_{sc}$: $A_f(x,y) = w\, M_f(x,y) + (1 - w)\, M_{sc}(x,y)$. The weight factor $w$ was set to 0.2346 across all the experiments, selected to optimize the fit between human and IVSN$_{size}$ saccade sizes. In a similar fashion to IVSN$_{fIOR}$ and to avoid overfitting with did cross-validation by fitting $M_{sc}$ separately for each experiment, using only a random subset of 7 out of the 15 subjects.

**Data analysis**. Psychophysics fixation analysis: We used the EDF2Mat function provided by the EyeLink software (SR Research, Canada) to automatically extract fixations. We clustered consecutive fixations that were within object bounding boxes of size 45 × 45 pixels for more than 50 ms. If fixation was not detected during the initial fixation window, the experimenter re-calibrated the eye tracker. The last trial before re-calibration and the first trial after calibration were excluded from analyses. In Experiment 1, we filtered out fixations falling outside the six object locations (13.7 ± 5.6% of the trials). Upon presentation of the search image, we considered the first fixation away from the center. We considered that a fixation had landed on the target object if it was within a square window centered on the target object. The window sizes were 45 × 45 for Experiment 1, 200 × 200 pixels for Experiment 2, and 100 × 100 pixels for Experiment 3. These values correspond to the mean widths and heights of all the ground truth bounding boxes for each dataset (Supplementary Figure 1). In Experiments 2 and 3, subjects had to click the target location with the mouse. The mouse click location had to fall on the window defining the target object location for the trial to be deemed successful. In 15.9 ± 4.9% of trials in Experiment 2 and 10.1 ± 7.0% of trials in Experiment 3, the initial mouse clicks were incorrect. If the location indicated by the mouse click was incorrect, subjects had to continue searching; otherwise, the trial was terminated. It should be noted that in several cases, subjects could fixate on the target object but not click the mouse, most likely because they were not consciously aware of finding the target despite the correct fixation (Supplementary Figure 12, see Discussion). As discussed above, for fair comparison with the models, we used an oracle version such that the target was considered to be found upon the first fixation on the target, except in Supplementary Figure 12.

Comparisons of fixation patterns: We evaluated the degree of within-subject consistency by comparing the fixations that subjects made during the first versus second presentation of a given target image and search image. We evaluated the degree of between-subject consistency by performing pairwise comparisons of the fixations that subjects made in response to the same target image and search image for all 15-choose-2 subject pairs. We compared the fixations of the IVSN model against each of the 15 subjects. We used the following metrics to compare fixations within subjects, between subjects and between subjects and the IVSN model: (1) we considered the cumulative accuracy as a function of the number of fixations to evaluate the overall search performance (Figs. 3e, 4e, and 5e); (2) we compared the number of fixations required to find the target on an image-by-image basis (Supplementary Figure 7); (3) we compared the spatiotemporal sequence of fixations on an image-by-image basis (Fig. 6, Supplementary Figure 8).

(1) Cumulative performance. We compute the probability distribution $p(n)$ that the subject or model finds the target in $n$ fixations. Figs. 3e, 4e, and 5e show the cumulative distribution of $p(n)$.

(2) Number of fixations to find the target. For each image, we plot the number of fixations required to find the target for S1 and S2 where S1 and S2 can be different repetitions of the same image (within-trial consistency), different subjects (between-trial consistency), or subject and model (model-subject consistency). This metric is reported in Supplementary Figure 7.

(3) Spatiotemporal dynamics of fixations on an image-by-image basis. We used the scanpath similarity score proposed by Borji et al.[27]. This measure takes into account both spatial and sequential order by aligning the scanpath

between two sequences. We used the implementation described in ref.[53]. Briefly, a mean-shift clustering for all human fixations was computed, and a unique character was assigned to each cluster center and corresponding fixations. The Needleman–Wunsch string match algorithm[54] was implemented to evaluate the similarity of a scanpath pair. In Supplementary Figure 8, we compare the entire sequences. In Fig. 6, we compare the first $x$ fixations as shown in the $x$-axis in the figure.

Statistical analyses: We used two-tailed t-tests when comparing two distributions and considered results to be statistically significant when $p < 0.01$. Because calculations of $p$ values tend to be inaccurate when the probabilities are extremely low, we reported all $p$ values less than $10^{-15}$ as $p < 10^{-15}$ (as opposed to reporting, for example, $p = 10^{-40}$); clearly none of the conclusions depend on this.

**Code availability**. All the source code is publicly available through the lab's GitHub repository: https://github.com/kreimanlab/VisualSearchZeroShot.

## Data availability

All the raw data are publicly available through the lab's GitHub repository: https://github.com/kreimanlab/VisualSearchZeroShot.

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

## Acknowledgements

We thank Farahnaz Wick for comments on the manuscript, and Suresh Krishna and Jeremy Wolfe for discussions. This work was supported by NIH (R01EY026025) and NSF grants (CCF-1231216) to G.K., by A*STAR JCO Grant 1335h00098 to J.H.L. and M.Z., A*STAR SERC Strategic Funds A1718g0048 to K.T.M.

## Author contributions

The behavioral experiments were designed by M.Z. and G.K. The computational model was designed by M.Z., G.K. and J.F. M.Z. collected the data, analyzed the data, and ran all the computational simulations. The manuscript was written by M.Z. and G.K. with comments from J.F., K.T.M., J.H.L., and Q.Z.

## Additional information

**Competing interests:** The authors declare no competing interests.

