## [Peer Review File · Nature Communications]

Reviewers' comments:

Reviewer #1 (Remarks to the Author):

This manuscript presents three visual search experiments in which observers look for targets that can vary in appearance from the canonical target image. The manuscript also presents a biologically-plausible computational model (Invariance Visual Search Network, or IVSN) to help understand how this task might be accomplished. Both humans and IVSN do better than a random searcher, or a set of alternative computational models (sliding window, Itti-Koch salience, random weight, pixel match).

From a technical standpoint, the manuscript is highly accomplished. The experiments are well-designed, and the three paradigms cover a useful range of search difficulty. The IVSN model is very sophisticated. I am also pleased that, while the manuscript is primarily concerned with overt eye movements as a measure of performance, the authors acknowledge the potential role of covert attentional shifts.

However, I found the paper somewhat frustrating. The authors conclude that "The large dissimilarity between how the targets were rendered in the search image and their appearance in the target image indicates that humans do not merely apply pixel-level template matching to find the target. These results suggest that the features guiding visual search must be invariant to target object transformations." (p. 16). First, did anybody think that we used pixel-level template matching? This ground has been covered decades ago in the object-recognition literature, and simple templates have been decisively rejected. This literature would seem to be highly relevant to the problem at hand, but is not mentioned at all. Of course, visual search and object recognition are not the same process, and it is certainly possible, even likely, that the features used for search are different from the features used for (fully attended) recognition, based on previous work (I note that Jeremy Wolfe is thanked in the acknowledgements section; he can give you plenty of examples). But the manuscript does not address this issue. Indeed, I don't see what we've learned about these transformation-invariant features that are used for search. We know that humans can successfully perform this kind of search, and so can the model. However, as the authors appropriately caution us, I wouldn't want to read too much into that. I'm not sure that the model is searching in the same way as the human observers.

A key difference between the model and human observers seems to be inhibition of return. On p.7, the authors write that, "The model had infinite inhibition-of-return and therefore never revisited the same location, by construction thus achieving 100% performance at 6 fixations." They go on to write, in the same paragraph, "the IVSN model

performance was similar to human behavior". Yet on p. 6, they note "At 6 fixations, the cumulative performance was below 100% (97±0.5%), since subjects tended to revisit the same locations, even when they were wrong.", and again on p. 9 "humans tended to revisit the same locations over and over again even though the target was not there". The only real acknowledgement of this in the paper comes on p. 20: "Yet, our model implements infinite inhibition of return as a simplifying assumption that could also be improved upon by including a memory decay function." (I would modestly point to Horowitz, T. S. (2006). Revisiting the variable memory model of visual search. *Visual Cognition*, 14(4), 668–684, as a starting point) But if humans and the model are using different sampling strategies, doesn't that substantially reduce the usefulness of cumulative performance plots (e.g., figures 3E and 4E) for the purposes of this paper? The shape of the plots (and the RT distributions) will be substantially shaped, not just by the way humans and IVSN deal with invariance, but by the sampling regimes.

Another difference between humans and the model is that humans perform very poorly on trials with more than a few saccades (p. 10: "Humans outperformed the model up to approximately fixation number 10,

but the model performed well above humans thereafter"). It seems like this should be telling us something important. What is it about the model that lets it outperform humans on these long-tail trials? What are the characteristics of the targets that humans fail to find, but the model does?

Finally, a major difficulty in using the IVSN model to gain insight into the way humans implement object invariance in search is that the model and the human are set somewhat different tasks, despite being given the same images. The IVSN does not actually have to decide whether or not a target is present (p. 29: "To isolate the search process from the verification process, we bypass the recognition question by using an 'oracle' system that decides whether the target is present or not. The oracle checks whether the selected fixation falls within the ground truth location, defined as the bounding box of the target object."). In other words, the model "found" the target once it moved its virtual fixation to the target. Humans, however, had to actually recognize the target. Indeed, the authors highlight in Figure 4C the behavior of human observers fixating the target, but moving on. Thus, the human visual search tasks did not "isolate the search process from the verification process", making the human and IVSN data qualitatively different. This is especially important when we consider, again, that the features that allow transformation-invariant object recognition and transformation-invariant visual search guidance are likely to be different.

On a different topic, I am concerned by the amount of behavioral data. If I read the Methods correctly, there were only 5 observers per experiment, contributing only 600 trials apiece. What was the rationale for this small sample size?

Todd S. Horowitz

Reviewer #2 (Remarks to the Author):

Review of "Finding any Waldo: invariant and efficient visual search" by Dr. Kreiman and colleagues, submitted for publication in Nature Communications, April 2018

Summary: An interesting study of how humans sequentially search for objects in visual scenes. Plus: a model of such search processes based on artificial neural networks (NNs).

Model: At a given time step, 2 parallel convolutional neural networks (CNNs) see: a small target image to be found in the scene (CNN1), and a patch of a visual scene which is partitioned into numerous such patches of size 224x224 pixels (CNN2).

Once a pre-wired "oracle" decides that the target is present in the current patch, the incremental search for the target through sequential saccade generation is stopped. (A variant of the system makes such decisions through an NN pre-trained on the ImageNet database.)

To generate candidates for the next fixation point, the target's internal representation in layer 31 of CNN1 is convolved with the representation of the current image patch in layer 30 of CNN2, to generate a 16x16 "attention map."

The most active element in the map corresponding to a location that has not been visited before determines the next fixation point.

In limited settings, the experimental results seem compatible with observations of human subjects.

Related work:

The authors cite selected, relatively recent, related papers. However, they do not compare at all to what's probably the first NN that learned sequential visual attention, and perhaps the most closely related one:

The reinforcement learning (RL) NN controller of ref [1] (1990) learns to control a fovea through sequences of saccades to find target objects in visual scenes, thus learning sequential attention. Like in the present submission, target images are provided through a separate input channel [1, Sec. 3.2] while the system shapes its stream of visual inputs through fovea-shifting actions.

[1] J. Schmidhuber and R. Huber. Learning to generate artificial fovea trajectories for target detection. *International Journal of Neural Systems*, 2(1 & 2):135–141, 1991. Based on TR FKI-128-90, TUM, 1990. More: <http://people.idsia.ch/~juergen/attentive.html>

In many ways, the RL model of 1990 [1] was actually more general than what the authors propose. It learned to compute sequences of fovea shifts or saccades, each saccade defined by a direction and magnitude, and sometimes even a rotation. Only at the end of a trajectory made up of, say, 20 saccades, the system got a reward in case the final visual input matched the target input. From this sparse information, the RL NN had to learn to generate useful intermediate saccades. Unlike in the present submission, there was neither a pre-defined attention map at every time step, nor a built-in inhibition of return, nor a pre-segmentation of the image into patches - the system had to learn everything by itself. (Of course, compute was a million times more expensive back then, that is, only toy experiments with black & white images were possible.)

It seems clear that the authors should clarify what's different to the early work, and why their more specialised approach should be preferred in certain settings.

Additional comments on related work:

line 651: mentions SVM classifier - so cite Vapnik et al's original work, or a survey, e.g.:

Schölkopf, Bernhard; Burges, Christopher J. C.; and Smola, Alexander J. (editors); *Advances in Kernel Methods: Support Vector Learning*, MIT Press, Cambridge, MA, 1999. ISBN 0-262-19416-3

line 813 ff: cite the first fast CNNs to win vision contests by Ciresan et al (2011) - the cited Alexnet etc was very similar but came later (2012) - see Schmidhuber's overview page: <http://people.idsia.ch/~juergen/computer-vision-contests-won-by-gpu-cnns.html>

line 845: mentions max-pooling - so cite its inventor (Weng et al, 1993) who added this to Fukushima's old basic CNN architecture (Neocognitron) of the 1970s:

Weng, J., Ahuja, N., and Huang, T. S. (1993). Learning recognition and segmentation of 3-D objects from 2-D images. *Proc. 4th Intl. Conf. Computer Vision*, Berlin, Germany, pp. 121-128.

Fukushima, K. (1979). Neural network model for a mechanism of pattern recognition unaffected by shift in position - Neocognitron. *Trans. IECE*, J62-A(10):658–665.

General comments: I think that the major claims of the paper will be of interest to others in the

community and the wider field, and might influence thinking in the field. Nevertheless, relations to previous work must be clarified. Revise and resubmit. I'd like to see the revised version again.

Reviewer #3 (Remarks to the Author):

This paper seeks to analyze the fixations humans perform to localize objects in images, and to build a computational model of the same.

The three sets of psychophysics experiments that the authors perform to analyze human performance are well designed, and the results are interesting and useful. However, I think the paper falls short of its stated aims in several ways:

- The authors emphasize the "zero-shot" nature of the problem: that the system has not seen the target before. While this is true for the machine vision system, it is not true for the human vision system, since all targets used in these experiments correspond to object classes humans have seen before. Therefore, it is possible that subjects are simply recognizing the target presented to them, and using their knowledge of the target class.

One possible way to avoid this might be to use fine-grained classes, such as different types of birds, since subjects may not know how to distinguish these a priori.

- The proposed computational model is too simplistic and doesn't capture many critical aspects. For example, the attention maps are produced just once, and information from subsequent fixations aren't used to update the attention map; whereas humans might use contextual information from previous fixations to decide where to look next. This might explain the low correlation between human fixations and model fixations in the second and third experiments.

- The proposed model is crucially dependent on the underlying network architecture, yet the impact of that is not evaluated. Are different architectures equally predictive of human performance? Does tapping into other layers of the network change the correlation? Do the many other object detection architectures work similarly?

Point-by-point responses to reviewers

General notes

Numbers in square brackets here denote line numbers in the revised manuscript.

Reviewers' questions are copied in red and small font here without editing. Responses are in black.

We note that we have added 10 more subjects per experiment. The conclusions remain consistent with the ones reported in the original submission (but all the psychophysics results figures have changed).

All the experimental data and open source code are publicly available in the lab's GitHub repository:

<https://github.com/kreimanlab/VisualSearchZeroShot>, and cited in the main text.

Reviewer 1

This manuscript presents three visual search experiments in which observers look for targets that can vary in appearance from the canonical target image. The manuscript also presents a biologically- plausible computational model (Invariance Visual Search Network , or IVSN) to help understand how this task might be accomplished. Both humans and IVSN do better than a random searcher, or a set of alternative computational models (sliding window, Itti- Koch salience, random weight, pixel match). From a technical standpoint, the manuscript is highly accomplished. The experiments are well- designed, and the three paradigms cover a useful range of search difficulty. The IVSN model is very sophisticated. I am also pleased that, while the manuscript is primarily concerned with overt eye movements as a measure of performance, the authors acknowledge the potential role of covert attentional shifts.

R1.4. On a different topic, I am concerned by the amount of behavioral data. If I read the Methods correctly, there were only 5 observers per experiment, contributing only 600 trials apiece. What was the rationale for this small sample size?

[We swapped the question order because this question impacts essentially all the figures]

As suggested by the reviewer, we have now added 10 more subjects per experiment (a total of 15 subjects per experiment). The conclusions remain consistent with the ones reported in the original submission (but all the figures and all the numbers have changed).

As a historical note of little relevance now, the reviewer correctly understood that we had 5 subjects per experiment. The rationale for this was based on a simulation that we ran by subsampling data from a previous study where we examined 20 subjects (Miconi et al, Cerebral Cortex 2016).

We also note that all the source code and behavioral data are publicly available from our GitHub link and lab web site.

R1.1 However, I found the paper somewhat frustrating. The authors conclude that "The large dissimilarity between how the targets were rendered in the search image and their appearance in the target image indicates that humans do not merely apply pixel- level template matching to find the target. These results suggest that the features guiding visual search must be invariant to target object transformations." (p. 16). First, did anybody think that we used pixel- level template matching? This ground has been covered decades ago in the object- recognition literature, and simple templates have been decisively rejected. This literature would seem to be highly relevant to the problem at hand, but is not mentioned at all. Of course, visual search and object recognition are not the same process, and it is certainly possible, even likely, that the features used for search are different from the features used for (fully attended) recognition, based on previous work (I note that Jeremy Wolfe is thanked in the acknowledgements section; he can give you plenty of examples). But the manuscript does not address this issue. Indeed, I don't see what we've learned about these transformation- invariant features that are used for search. We know that humans can successfully perform this kind of search, and so can the model. However, as the authors appropriately caution us, I wouldn't want to read too much into that. I'm not sure that the model is searching in the same way as the human observers.

The reviewer makes multiple important and correct points here.

[R1.1.a] Template matching.

Expanding on what the reviewer pointed out, and writing fast and loose, visual recognition can be thought of primarily as a "bottom-up" process and eye movement guidance during visual search as a "top-down" process (this is of course a major oversimplification: object recognition also uses top-down

signals and eye movement guidance during visual search is also heavily influenced by bottom-up processes). The fact that template matching is a very bad model for visual recognition does not imply that template matching is also a bad model for visual search. We now explicitly point out in the text that template matching is a very poor model for visual recognition [217-218].

Even though template matching cannot address the problem of invariance in visual recognition, it continues to be used to solve computer vision problems (e.g.¹⁻⁵). Given that the problem of invariant visual search is related to the problem of object detection in the computer vision literature, we compare the results against template matching (as well as many other models, **Fig. S4**, **new Fig. S11**, **new Fig. S13**, **new Fig. S14**). As expected, template matching is not robust to invariant transformations and its search efficiency is inferior to humans and to IVSN (**Fig. S4**) [217-219, 296-300, 585-587, 595-596].

The reviewer may well consider template matching to be a “straw man hypothesis” and we would agree. In fact, we would go further and argue that all the “null” models in **Fig. S4** are straw man models (random eye movements, random weights, bottom-up saliency, template matching). That is why we refer to them as “null” models [157, 191, 207, 221, 257, 259, 281, 292, 346, 359, 363, 586]

We are certainly *not* trying to argue that template matching is the standard in the field (in the same way that we are *not* trying to argue that random eye movements are the standard in the field either). We do note, however, that many visual search experiments in the literature are based on scenarios where the target is exactly identical in the target image and the search image. These search tasks *might* be solved by template matching (but we agree with the reviewer that it seems highly unlikely that subjects use a template matching strategy even in those cases). We find it useful and instructive to compare results against template matching (and other null models), but we certainly share the reviewer’s intuition that these are not good models.

To directly evaluate visual search under realistic conditions where template matching cannot solve the problem, the appearance of the target object in the target image was different from the appearance of the target objects in the search image in all the experiments; this is what we refer to as the *invariance problem in visual search*. To the best of our knowledge, (and to the best of Prof. Wolfe’s knowledge), invariance in visual search has not been systematically studied.

[R1.1b] Are the features used for visual recognition the same features used for visual search?

The short answer is that we do not know. There are four *indirect* arguments of plausibility that suggest that visual recognition and visual search *may* share similar features:

(i) Directing attention to a particular location depends both on bottom-up and top-down signals. Rapid parallel bottom-up signals can be approximately thought of as processes required for visual recognition and can have a large impact on attentional allocation⁶⁻¹¹.

(ii) *All* the brain regions along the ventral visual stream required for object recognition are modulated by attentional mechanisms (e.g.¹²⁻¹⁶). At the physiological level, top-down attentional mechanisms act on the same neurons that subserve visual recognition.

(iii) There is a rich psychophysics literature documenting features that are important for visual search (reviewed in⁹). These features include color^{17,18}, orientation^{8,19}, spatial frequency^{11,17}, curvature and depth¹⁹. *All* of these features have also been well studied and proven to be useful for object recognition. The IVSN model uses all of these features.

(iv) Several computational models have suggested a close relationship between object recognition and attention. These models leverage on the features at the attended location for improving their object recognition performance^{10,17,19-24}.

These studies may not convince the reviewer, who is right in raising this interesting question, and we emphasize that these are plausibility arguments rather than a mathematical proof that recognition and search features are the same. We added a brief sentence in the discussion [643-646]

to highlight that this is an important and interesting question that will require further investigation. We think that it is parsimonious to consider that the answer may be yes (i.e., that object recognition features are used in visual search). The proposed IVSN model can search for objects using features that were learned for visual recognition (see further discussion in R1.1c).

R1.1c. The reviewer incisively points out: “I’m not sure that the model is searching in the same way as the human observers.”

We completely agree with this statement as well. We have now pasted this statement, almost verbatim, in the text [690-692]. The model does *not* perfectly match human behavior, we note this repeatedly throughout the manuscript (e.g., **Fig. 4E, 5E, 6**) and we devote a whole section in the discussion [698-741] to highlight several aspects of the proposed model that deserve further scrutiny.

At the risk of providing an answer that is too philosophical, let us assume for a moment that we had a model that can perfectly predict human behavior, not only on average (**Fig. 3, 4, 5**) but also at the image-by-image level (**Figs. S7**), and can also explain the spatiotemporal sequence of eye movements (**Fig. 6, S8**). As the reviewer correctly points out, even such a model may not be searching in the same way as the human observers.

The IVSN model is (coarsely) based on neuroanatomical and neurophysiological constraints (**Fig. 2**) [86-100, 175-180, 553-554, 961-964, 964-968, 977-978], and we cautiously think that it provides a reasonable first order approximation to eye movements during natural visual search [113-114, 742-743]. Yet, it is very clear that there is a lot of ingredients missing and we are very far from having a model that can search the way humans do. There are several important aspects of the model that merit improvement and further work [698-741].

R1.2

A key difference between the model and human observers seems to be inhibition of return. On p.7, the authors write that, "The model had infinite inhibition- of- return and therefore never revisited the same location, by construction thus achieving 100% performance at 6 fixations." They go on to write, in the same paragraph, "the IVSN model performance was similar to human behavior". Yet on p. 6, they note "At 6 fixations, the cumulative performance was below 100% (97±0.5%), since subjects tended to revisit the same locations, even when they were

wrong.", and again on p. 9 "humans tended to revisit the same locations over and over again even though the target was not there". The only real acknowledgement of this in the paper comes on p. 20: "Yet, our model implements infinite inhibition of return as a simplifying assumption that could also be improved upon by including a memory decay function." (I would modestly point to Horowitz, T. S. (2006). Revisiting the variable memory model of visual search. *Visual Cognition*, 14 point) But if humans and the model are using different sampling strategies, doesn't that substantially reduce the usefulness of cumulative performance plots (e.g., figures 3E and 4E) for the purposes of this paper? The shape of the plots (and the RT distributions) will be substantially shaped, not just by the way humans and IVSN deal with invariance, but by the way humans and IVSN deal with invariance, but by the sampling regimes.

We agree again with the comments made by the reviewer here. In a schematic fashion, the decision to move the eyes to a particular location during visual search depends on: (i) the target features and the search image features, (ii) memory of previously visited locations, (iii) a decision mechanism that recognizes whether the target was found or to continue searching, (iv) other constraints such as the eye muscles precluding from making very large saccades. As a simplifying working hypothesis, we assume that these processes are independent and we focus our study on item (i) by presenting the default IVSN model with infinite inhibition-of-return (ignoring (ii)), with an oracle that determines whether the target was found or not (ignoring (iii)), and with an approximately uniform distribution of saccade sizes (ignoring (iv)).

We further present initial sketches of variations of the model that relax those assumptions: IVSN_{recognition} (**new Fig. S11A-C**) presents an initial model variation that addresses (iii) by determining whether the target is present or not based on a simple object recognition classifier^{25,26}.

IVSN_{distance} (**new Fig. S11G-L**) presents an initial model variation that addresses (iv) where the saccade sizes are constrained by the distribution of saccade sizes that humans make

IVSN_{fIOR} (**new Fig. S11D-F**) presents an initial model variation that addresses (ii) that shows finite inhibition of return based on the work of Horowitz²⁷.

As the reviewer knows very well, humans tend to revisit the same locations, i.e. they do *not* have infinite inhibition of return. We had pointed this out in a rather vague and anecdotal manner in the original submission. Following the suggestion from the reviewer, we implemented a finite inhibition of return (fIOR) mechanism inspired by the work suggested by the reviewer²⁷. We note that in Experiments 2 and 3, we do not have a list of isolated objects and thus we cannot directly implement a memory function that is based on objects. Instead, the memory function was obtained by empirically fitting the human behavioral data calculating the probability of revisiting the same place after a number of fixations. In the IVSN_{fIOR} model, the attention map gets modulated by this memory function across fixations. As the reviewer predicted, the IVSN_{fIOR} is closer to human performance (**new Fig. S11D-F**).

The reviewer criticizes the cumulative performance plots but we think that these plots provide useful information to compare human performance and different models, and cumulative distributions provide richer information than merely reporting average number of fixations. We further note that we present a rather extensive set of metrics to compare the model and humans: cumulative performance (**Figs. 3E, 4E, 5E** and multiple supplementary figures), image-by-image comparison of the number of fixations (**Fig. S7**), image-by-image comparisons in the spatiotemporal sequence of fixations (**Figures 6, S8**), distribution of saccade sizes (**new Fig. S11G-I**), overall distribution of fixation locations (**Fig. S1**), target found times (**Figures S2B, D, F**), reaction times for the first 6 saccades (**Figs. 3D, 4D, 5D, S2A, C, E**), distance to target for the last 6 fixations (**new Figs. S2G-L**). We also implemented several other comparisons including (location of the first fixation, feature similarity between model fixations and human fixations, euclidian distance between fixation locations); we opted to omit all these other comparisons from the manuscript because of the rather large number of figures and supplementary figures. We would certainly welcome suggestions as to other and better metrics for comparing models and human behavior.

R1.3 Finally, a major difficulty in using the IVSN model to gain insight into the way humans implement object invariance in search is that the model and the human are set somewhat different tasks, despite being given the same images. The IVSN does not actually have to decide whether or not a target is present (p. 29: "To isolate the search process from the verification process, we bypass the recognition question by using an 'oracle' system that decides whether the target is present or not. The oracle checks whether the selected fixation falls within the ground truth location, defined as the bounding box of the target object."). In other words, the model "found" the target once it moved its virtual fixation to the target. Humans, however, had to actually recognize the target. Indeed, the authors highlight in Fig. 4C the behavior of human observers fixating the target, but moving on. Thus, the human visual search tasks did not "isolate the search process from the verification process", making the human and IVSN data qualitatively different. This is especially important when we consider, again, that the features that allow transformation- invariant object recognition and transformation- invariant visual search guidance are likely to be different.

This is a very good point and we realize that we were being unfair to humans. We have now re-analyzed all the data by using an "oracle" also for human visual search (except for **new Fig. S12** where we compare human performance with and without oracle recognition). Once subjects fixate on the target, the target is considered to be found, therefore implementing the same criterion used for the models [1039-1042].

As we note in the text [474-496, 1103-1125], and the reviewer knows quite well, humans can fixate on the target, not realize that they have found the target and continue the search process by

moving their eyes elsewhere. An example of this behavior is shown in subject 5 in **Fig. 4**. This phenomenon is quantified in **new Figures S12A-B** and its effect on search performance is shown in **new Fig. S12C-D**. We provide an initial sketch of a model that implements recognition (IVSN_{recognition}, **new Fig. S11A-C**). To focus on the guidance aspects of eye movements, all the other figures use an oracle version both for humans and models, as suggested by the reviewer.

Reviewer 2

Summary: An interesting study of how humans sequentially search for objects in visual scenes. Plus: a model of such search processes based on artificial neural networks (NNs).

Model: At a given time step, 2 parallel convolutional neural networks (CNNs) see: a small target image to be found in the scene (CNN1), and a patch of a visual scene which is partitioned into numerous such patches of size 224x224 pixels (CNN2). Once a pre-wired "oracle" decides that the target is present in the current patch, the incremental search for the target through sequential saccade generation is stopped. (A variant of the system makes such decisions through an NN pre-trained on the ImageNet database.) To generate candidates for the next fixation point, the target's internal representation in layer 31 of CNN1 is convolved with the representation of the current image patch in layer 30 of CNN2, to generate a 16x16 "attention map." The most active element in the map corresponding to a location that has not been visited before determines the next fixation point. In limited settings, the experimental results seem compatible with observations of human subjects.

R2.1 Related work:

The authors cite selected, relatively recent, related papers. However, they do not compare at all to what's probably the first NN that learned sequential visual attention, and perhaps the most closely related one:

The reinforcement learning (RL) NN controller of ref [1] (1990) learns to control a fovea through sequences of saccades to find target objects in visual scenes, thus learning sequential attention. Like in the present submission, target images are provided through a separate input channel [1, Sec. 3.2] while the system shapes its stream of visual inputs through fovea-shifting actions.

[1] J. Schmidhuber and R. Huber. Learning to generate artificial fovea trajectories for target detection. *International Journal of Neural Systems*, 2(1 & 2):135–141, 1991. Based on TR FKI- 128- 90, TUM, 1990. More: <http://people.idsia.ch/~juergen/attentive.html>

In many ways, the RL model of 1990 [1] was actually more general than what the authors propose. It learned to compute sequences of fovea shifts or saccades, each saccade defined by a direction and magnitude, and sometimes even a rotation. Only at the end of a trajectory made up of, say, 20 saccades, the system got a reward in case the final visual input matched the target input. From this sparse information, the RL NN had to learn to generate useful intermediate saccades. Unlike in the present submission, there was neither a pre-defined attention map at every time step, nor a built-in inhibition of return, nor a pre-segmentation of the image into patches - the system had to learn everything by itself. (Of course, compute was a million times more expensive back then, that is, only toy experiments with black & white images were possible.) It seems clear that the authors should clarify what's different to the early work, and why their more specialised approach should be preferred in certain settings.

We thank the reviewer for pointing us towards this interesting work, which we have now cited. This is a very original effort where the authors propose a computational model which learns the policy of generating sequences of fovea shifts via reinforcement learning (reward only at goal).

There are several important differences with the work that we present here: (1) In the current work, we directly and systematically compared human invariant visual search eye movement behavior with the proposed model as well as multiple variations of the proposed model. (2) As pointed out by the reviewer, it is not clear whether the 1991 model of Schmidhuber and Huber can generalize to perform

invariant visual search with zero-shot training in complex natural scenes. In our study, we demonstrate that IVSN could generalize across many natural visual search tasks, show invariance to shape transformation, without extensive training for specific object classes. Of course, as the reviewer points out, this is not a fair comparison because we are writing the paper 27 years afterwards, with much heavier computational power at our disposal. (3) We are interested in studying zero-shot visual search problems which humans solve in daily life. Given a novel target exemplar, humans could search for novel objects with zero training on this novel object search task. In the main text, IVSN could generalize to search for novel objects with zero training. It is not clear to us whether the 1991 model could achieve this. (4) As the reviewer points out, a particularly interesting and impressive aspect of the 1991 model is that it can learn to execute eye movements purely via reinforcement learning signals with a delayed reward policy. Our model for the attention map generation is end-to-end trainable. In this study, we provided a demonstration of top-down feature biasing by taking the maximum on the attention map resulting in a sequence of fixations; in the future work, IVSN can be improved by training or fine-tuning via reinforcement learning as proposed in the 1991 study for various search tasks depending on the applications. We mention this now in the text in [722-727].

R2.2 Additional comments on related work:

line 651: mentions SVM classifier - so cite Vapnik et al's original work, or a survey, e.g.:

Schölkopf, Bernhard; Burges, Christopher J. C.; and Smola, Alexander J. (editors); *Advances in Kernel Methods: Support Vector Learning*, MIT Press, Cambridge, MA, 1999. ISBN 0- 262- 19416- 3

We cited this work as suggested [881]

R2.3

line 813 ff: cite the first fast CNNs to win vision contests by Ciresan et al (2011) - the cited Alexnet etc was very similar but came later (2012) - see Schmidhuber's overview page:

<http://people.idsia.ch/~juergen/computer-vision-contests-won-by-gpu-cnns.html>

We cited this work as suggested [52, 977]

R2.4 line 845: mentions max- pooling - so cite its inventor (Weng et al, 1993) who added this to Fukushima's old basic CNN architecture (Neocognitron) of the 1970s:

Weng, J., Ahuja, N., and Huang, T. S. (1993). Learning recognition and segmentation of 3- D objects from 2- D images. Proc. 4th Intl. Conf. Computer Vision, Berlin, Germany, pp. 121- 128.

Fukushima, K. (1979). Neural network model for a mechanism of pattern recognition unaffected by shift in position - Neocognitron. Trans. IECE, J62- A(10):658-665.

We cited these papers as suggested [52, 977]

General comments: I think that the major claims of the paper will be of interest to others in the community and the wider field, and might influence thinking in the field. Nevertheless, relations to previous work must be clarified. Revise and resubmit. I'd like to see the revised version again.

Reviewer 3

This paper seeks to analyze the fixations humans perform to localize objects in images, and to build a computational model of the same. The three sets of psychophysics experiments that the authors perform to analyze human performance are well designed, and the results are interesting and useful. However, I think the paper falls short of its stated aims in several ways:

R3.1 The authors emphasize the "zero-shot" nature of the problem: that the system has not seen the target before. While this is true for the machine vision system, it is not true for the human vision system, since all targets used in these experiments correspond to object classes humans have seen before. Therefore, it is possible that subjects are simply recognizing the target presented to them, and using their knowledge of the target class. One possible way to avoid this might be to use fine-grained classes, such as different types of birds, since subjects may not know how to distinguish these a priori.

This is an excellent point and we have conducted a new experiment to address it (**new Fig. S10**) [367-391, 931-955]. For this new experiment, we used a set of novel objects that subjects had never seen before (examples of these novel objects are shown in **new Fig. S10A**). These types of objects have been used in many other psychophysical experiments to evaluate the role of experience in recognition (e.g.²⁸⁻³¹). A critical consideration in visual search experiments is the degree of similarity between the target and distractors (e.g., visual search can be extremely challenging if the target differs from the distractors in only one pixel whereas visual search can be extremely easy if the task involves searching for a colorful Waldo among a set of black squares as distractors). Importantly, we matched the similarity between targets and distractors for known objects and the novel objects (**new Fig. S10C**). The results show that humans are able to perform visual search for this set of novel objects ("zero shot search") despite changes in the object appearance between the target image and search image ("invariance"), as shown in **new Fig. S10**.

While we liked the suggestion of using different types of birds, and we may do this in future experiments, we were concerned that it would be more challenging to carefully match the level of target/distractor similarity with different types of birds and we were also concerned that subjects might have different levels of expertise with birds.

R3.2 The proposed computational model is too simplistic and doesn't capture many critical aspects. For example, the attention maps are produced just once, and information from subsequent fixations aren't used to update the attention map; whereas humans might use contextual information from previous fixations to decide where to look next. This might explain the low correlation between human fixations and model fixations in the second and third experiments.

Indeed, the reviewer correctly points out that there are many simplifying assumptions in our model. We explicitly mention these assumptions in the text [193-194, 284, 474-475, 498-499, 520-521, 535-537, 553-554, 958-975] and we emphasize several simplifications in the discussion as well [689-738]. One simplification is that the IVSN model essentially does not use prior fixations to update the attention map ("essentially" because inhibition of return does use the prior saccade information by inhibiting it). We have introduced two variations of the model that incorporate information from previous fixations: IVSN_{size} constrains the size of the model saccades to approximate the distribution of human saccade sizes. The attentional map is weighted by the prior fixation in such a way that very large saccades are less likely (**new Fig. S11G-L**).

IVSN_{IOR} implements a finite inhibition-of-return mechanism whereas the model remembers previous fixations and weighs that information into the attentional map (**new Fig. S11D-F**).

When the reviewer mentions "contextual information", he/she may be referring to high-level understanding of the visual scene, which is certainly important and quite fascinating. For example, when searching for the car keys, humans know that it is highly unlikely that the keys are glued to the

ceiling and that it is far more likely that they might be resting on a desk. The IVSN model does *not* have this type of knowledge. In its most general form, this type of contextual information is a rather big and fascinating topic, which we hope to address in future work. We now explicitly mention this point in the text that [728-738]

In terms of the correlation between models and humans, we note that an upper bound to how well a model can explain human eye movement behavior in single trials is given by the between-subject and within-subject correlations. We note that these correlations are also relatively low, in other words, given the same target image and the same search image in two different trials, the same subject may use a different eye movement path to locate the target in a different trial (within subject comparisons) and another subject may also use a different path (between subject comparison). We also note that we have updated **Fig. 6** while moving the previous version to **Fig. S8**. The original version compared the *entire* fixation sequences, including cases where the two sequence lengths were very different. It is not very clear what the comparison between spatiotemporal sequences represents when the lengths difference is very large (e.g., in what sense is a sequence of length 2 similar or not to a sequence of length 40?). Instead, the new version of Fig. 6 directly compares the first x fixations, therefore evaluating the spatiotemporal patterns for sequences of the same length. Irrespective of which comparison metric we use (number of fixations, different spatiotemporal sequence metrics), the within-subject sequence similarity is higher than the between-subject one, which is in turn higher than the IVSN-human similarity (**Fig. 6, S7, S8**) [403-434, 451-456].

R3.3 The proposed model is crucially dependent on the underlying network architecture, yet the impact of that is not evaluated. Are different architectures equally predictive of human performance? Does tapping into other layers of the network change the correlation? Do the many other object detection architectures work similarly?

This is also a very interesting question. We answer the different parts separately.

R3.3a. “Does tapping into other layers of the network change the correlation?”

We have now added a series of alternative models where top-down modulation is applied at different layers of the network as suggested (**new Fig. S13**). Essentially, all of these model variations performed well above chance. In general, higher-layer features provide better performance and a better approximation to human behavior [535-551]. However, the relationship between performance and layer number is not monotonic: in some cases, intermediate layer features showed the highest performance (e.g., **new Fig. S13A**). We note that we did not try to fine tune the model or choose specific layers to optimize performance or match human behavior; there is certainly ample room to improve the model via training using a loss function based on performance and/or human behavior [725-727].

R3.3b. “Are different architectures equally predictive of human performance?”

This question is certainly very important but somewhat ill-defined and open-ended without specifying which other architectures the reviewer is referring to (see also R3.3.c below).

Although the reviewer probably does not consider the following models to represent “different architectures”, we evaluated multiple alternative “null” models: a random eye movement model (**Figs. 3, 4 and 5**), a sliding window model (**Fig. S4**), a random weights model (**Fig. S4**), a bottom-up saliency model (**Fig. S4**), and a template matching model (**Fig. S4**).

The reviewer may or may not consider the following models to represent “different architectures”. We evaluated four important extensions to the model: (i) IVSN-like models with top-down modulation applied to different layers (**R3.3a, new Fig. S13**); (ii) the IVSN_{recog} model, which includes a recognition component (**new Fig. S11A-C**); (iii) the IVSN_{fIOR} model, with finite inhibition of return (**new Fig. S11D-F**); (iv) the IVSN_{size} model constrained by the distribution of saccade sizes (**new Fig. S11G-L**).

In addition, we have introduced variations of the model using different “ventral visual cortex” modules (**new Fig. S14**). Specifically, we replaced the “ventral visual cortex” module in **Fig. 2** (which was based on VGG16²⁵) by several other plausible “ventral visual cortex” architectures: AlexNet³², ResNet³³, and FastRCNN³⁴.

This is admittedly *not* an exhaustive list of possible architectures and we would welcome specific suggestions of other architectures and other models that should be compared here.

R3.3c. “Do the many other object detection architectures work similarly?”

Again, this question is also certainly very important and somewhat ill-defined without specifying which object detection architectures the reviewer is referring to.

In R3.3b, we refer to the results obtained using different possible ventral visual cortex architectures. The reviewer might be referring to object detection architectures that aim to localize an object category in an image (e.g.³⁴⁻³⁸). We note that those object detection architectures depend heavily on extensive training with the chosen categories. Taking the popular and successful YOLO architecture as an example, YOLO thrives in setting a bounding box around a bicycle after *being trained with a large number of bicycles*. To the best of our knowledge, YOLO (without any modifications or training) cannot find Waldo. IVSN is based on “transfer learning” whereby we use a network that was trained for object categorization with a limited set of categories (VGG16) and we incorporate it into our model to measure performance in visual search with never-seen-before categories without any additional training. We expect that an object detection architecture (extensively trained with the categories of objects that the model has to search for) will perform quite well (as demonstrated in the references above). However, we *cannot* take an object detection architecture that was trained on ImageNet categories (e.g. trained to localize cars, kites and trees among other categories) and evaluate (without re-training) whether that architecture is or is not able to search for categories that it was not trained on (e.g. non ImageNet categories in Experiment 2, Waldo in Experiment 3, or the novel objects in Fig. S10).

References

- 1 Kawanishi, T., et al. in *IEEE International Conference on Pattern Recognition (ICPR)* Vol. 3 (2004).
- 2 Schonfeld, D. in *IEEE Transactions on Image Processing* Vol. 9 945-949 (2000).
- 3 Gharavi-Alkhansari, M. in *IEEE Transactions on Image Processing* Vol. 10 526-533 (2001).
- 4 Guskov, I. in *Computer Vision and Pattern Recognition (CVPR)* Vol. 1 (IEEE Computer Science 2006).
- 5 Dufour, R., Miller, E. & Galatsanos, N. in *IEEE Transactions on Image Processing* 1385-1396 (IEEE).
- 6 Nakayama, K. & Silverman, G. H. Serial and parallel processing of visual feature conjunctions. *Nature* **320**, 264-265, doi:10.1038/320264a0 (1986).
- 7 Pashler, H. Detecting conjunctions of color and form: reassessing the serial search hypothesis. *Perception & psychophysics* **41**, 191-201 (1987).
- 8 Wolfe, J. M., Cave, K. R. & Franzel, S. L. Guided search: an alternative to the feature integration model for visual search. *J Exp Psychol Hum Percept Perform* **15**, 419-433 (1989).
- 9 Wolfe, J. M. & Horowitz, T. S. Five factors that guide attention in visual search. *Nature Human Behaviour* **1**, 0058 (2017).
- 10 Itti, L., Koch, C. & Niebur, E. A model of saliency- based visual attention for rapid scene analysis. *IEEE Transactions on pattern analysis and machine intelligence* **20**, 1254-1259 (1998).

- 11 Duncan, J. & Humphreys, G. W. Visual search and stimulus similarity. *Psychol Rev* **96**, 433-458 (1989).
- 12 Moran, J. & Desimone, R. Selective attention gates visual processing in the extrastriate cortex. *Science* **229**, 782-784 (1985).
- 13 Luck, S. J., Chelazzi, L., Hillyard, S. A. & Desimone, R. Neural mechanisms of spatial selective attention in areas V1, V2, and V4 of macaque visual cortex. *J Neurophysiol* **77**, 24-42, doi:10.1152/jn.1997.77.1.24 (1997).
- 14 Chelazzi, L., Duncan, J., Miller, E. K. & Desimone, R. Responses of neurons in inferior temporal cortex during memory-guided visual search. *J Neurophysiol* **80**, 2918-2940, doi:10.1152/jn.1998.80.6.2918 (1998).
- 15 Motter, B. C. Focal attention produces spatially selective processing in visual cortical areas V1, V2, and V4 in the presence of competing stimuli. *J Neurophysiol* **70**, 909-919, doi:10.1152/jn.1993.70.3.909 (1993).
- 16 Reynolds, J. H., Pasternak, T. & Desimone, R. Attention increases sensitivity of V4 neurons. *Neuron* **26**, 703-714 (2000).
- 17 Treisman, A. M. & Gelade, G. A feature-integration theory of attention. *Cognitive psychology* **12**, 97-136 (1980).
- 18 Humphreys, G. W., Quinlan, P. T. & Riddoch, M. J. Grouping processes in visual search: effects with single- and combined-feature targets. *J Exp Psychol Gen* **118**, 258-279 (1989).
- 19 Wolfe, J. M. Guided Search 4.0. *Integrated models of cognitive systems*, 99-119 (2007).
- 20 Koch, C. & Ullman, S. Shifts in selective visual attention: towards the underlying neural circuitry. *Matters of Intelligence*, 115-141 (1987).
- 21 Deco, G. & Rolls, E. T. *Computational Neuroscience of Vision*. (Oxford University Press, 2004).
- 22 Tsotsos, J. Analyzing Vision at the Complexity Level. *Behavioral and Brain Sciences* **13-3**, 423-445 (1990).
- 23 Tsotsos, J. *et al.* Modeling visual attention via selective tuning. *Artificial Intelligence* **78**, 507-545 (1995).
- 24 Grossberg, S., Mingolla, E. & Ross, W. D. A neural theory of attentive visual search: interactions of boundary, surface, spatial, and object representations. *Psychol Rev* **101**, 470-489 (1994).
- 25 Simonyan, K. & Zisserman, A. Very deep convolutional networks for large-scale image recognition. *arXiv*, 1409.1556 (2014).
- 26 Serre, T. *et al.* A quantitative theory of immediate visual recognition. *Progress in brain research* **165**, 33-56, doi:10.1016/S0079-6123(06)65004-8 (2007).
- 27 Horowitz, T. S. Revisiting the variable memory model of visual search. *Visual cognition* **14**, 668-684 (2006).
- 28 Gauthier, I. & Tarr, M. J. Becoming a "Greeble" expert: exploring mechanisms for face recognition. *Vision Research* **37**, 1673-1682 (1997).
- 29 Duchaine, B. C., Dingle, K., Butterworth, E. & Nakayama, K. Normal greeble learning in a severe case of developmental prosopagnosia. *Neuron* **43**, 469-473, doi:10.1016/j.neuron.2004.08.006 (2004).
- 30 Barry, T. J., Griffith, J. W., De Rossi, S. & Hermans, D. Meet the Fribbles: novel stimuli for use within behavioural research. *Front Psychol* **5**, 103, doi:10.3389/fpsyg.2014.00103 (2014).
- 31 Hayward, W. G. & Tarr, M. J. Testing conditions for viewpoint invariance in object recognition. *J Exp Psychol Hum Percept Perform* **23**, 1511-1521 (1997).
- 32 Krizhevsky, A., Sutskever, I. & Hinton, G. E. Imagenet classification with deep convolutional neural networks. *Advances in Neural Information Processing Systems*, 1097-1105 (2012).
- 33 He, K., Zhang, X., Ren, S. & Sun, J. in *Proceedings of the IEEE conference on computer vision and pattern recognition*. 770-778.
- 34 Ren, S., He, K., Girshick, R. & Sun, J. Faster r-cnn: Towards real-time object detection with region proposal networks. *Advances in Neural Information Processing Systems* 91-99 (2015).
- 35 Girshick, R., Donahue, J., Darrell, T. & Malik, J. in *Proceedings of the IEEE Conference on Computer Vision and Pattern Recognition* 580-587 (2014).

- 36 Erhan, D., Szegedy, C., Toshev, A. & Anguelov, D. Scalable object detection using deep neural networks. *Proceedings of the IEEE Conference on Computer Vision and Pattern Recognition*, 2147-2154 (2014).
- 37 Szegedy, C., Toshev, A. & Erhan, D. in *Advances in Neural Information Processing Systems*.
- 38 Redmon, J., Divvala, S., Girshick, R. & Farhadi, A. in *Proceedings of the IEEE conference on computer vision and pattern recognition*.

REVIEWERS' COMMENTS:

Reviewer #1 (Remarks to the Author):

The authors have done an impressively thorough job of responding to my critique, including tripling the sample size and adding a new version of the model with finite IOR. I had prepared a longer response, but most of the comments were variations on "That's great!" or "Fair point". I am quite satisfied with the revisions and find the ms appropriate for publication in Nature Communications. I have one remaining quibble: It's great that the authors have added ten more human observers. However, the rationale for choosing this sample size should be in the method section, so that the reader can judge for themselves.

Reviewer #2 (Remarks to the Author):

I think the changes are acceptable.

Reviewer #3 (Remarks to the Author):

This review should be seen as updates to my previous review.

1. Zero-shot search: In my previous review I had commented that the claim that this was zero-shot recognition did not hold watersince humans may have seen the objects before. The authors have responded by conducting a new experiment with unseen objects. I appreciate this new experiment and I think this issue has been addressed appropriately.

2. Simplistic architectures: I had suggested that the architecture of IVSN presented is too simplistic. This comment has not been addressed appropriately. I want to reiterate: when making saccades, presumably each saccade reveals new information to the system (above and beyond just the presence of the object at the saccade location), which should help the system decide upon the next saccade. To ignore this new information is to discard quite a lot, and I don't see a cogent argument that the human visual system is discarding vital information in this way. Perhaps the big difference between the human performance and computational models in Figure 6 might be because of this.

I am willing to accept the paper even if the proposed computational model does not include or evaluate this, but I do think not having this feature or evaluating its contribution dramatically reduces the impact of the paper.

3. Impact of model architecture: I appreciate the fact that the authors have also tried using the VGG, ResNet and AlexNet architectures. That is exactly what I wanted. Please include all architectures (and indeed all IVSN variants including those in the supplementary) in Figure 6.

Final recommendation: Accept with minor changes (see 3. above)

Response to Reviewer's comments

Reviewer #1 (Remarks to the Author):

The authors have done an impressively thorough job of responding to my critique, including tripling the sample size and adding a new version of the model with finite IOR. I had prepared a longer response, but most of the comments were variations on "That's great!" or "Fair point". I am quite satisfied with the revisions and find the ms appropriate for publication in Nature Communications. I have one remaining quibble: It's great that the authors have added ten more human observers. However, the rationale for choosing this sample size should be in the method section, so that the reader can judge for themselves.

We have added this rationale to the Methods section as suggested.

Reviewer #2 (Remarks to the Author):

I think the changes are acceptable.

Reviewer #3 (Remarks to the Author):

This review should be seen as updates to my previous review.

1. Zero-shot search: In my previous review I had commented that the claim that this was zero-shot recognition did not hold watersince humans may have seen the objects before. The authors have responded by conducting a new experiment with unseen objects. I appreciate this new experiment and I think this issue has been addressed appropriately.

2. Simplistic architectures: I had suggested that the architecture of IVSN presented is too simplistic. This comment has not been addressed appropriately. I want to reiterate: when making saccades, presumably each saccade reveals new information to the system (above and beyond just the presence of the object at the saccade location), which should help the system decide upon the next saccade. To ignore this new information is to discard quite a lot, and I don't see a cogent argument that the human visual system is discarding vital information in this way. Perhaps the big difference between the human performance and computational models in Figure 6 might be because of this.

I am willing to accept the paper even if the proposed computational model does not include or evaluate this, but I do think not having this feature or evaluating its contribution dramatically reduces the impact of the paper.

We agree with the general theme in this comment, even though the reviewer does not explicitly state what type of information is shared across saccades. The current version of the model does incorporate information about previous saccades (e.g. inhibition of return and distance constraints). However, the reviewer is probably referring to additional visual information that could be carried from previous fixations and could well help plan the next saccade. We now make this point explicitly in the Discussion.

3. Impact of model architecture: I appreciate the fact that the authors have also tried using the VGG, ResNet and AlexNet architectures. That is exactly what I wanted. Please include all architectures (and indeed all IVSN variants including those in the supplementary) in Figure 6.

We have incorporated all models into Figure 6 as suggested.